# LongWriter-Zero: Mastering Ultra-Long Text Generation via Reinforcement Learning

**Yuhao Wu[*,1], Yushi Bai[*,1], Zhiqiang Hu[1], Roy Ka-Wei Lee[†,1], Juanzi Li[†,2]**

[1]Singapore University of Technology and Design, Singapore
[2]Tsinghua University, Beijing, China
{wu_yuhao,zhiqiang_hu}@mymail.sutd.edu.sg,
bys22@mails.tsinghua.edu.cn
roy_lee@sutd.edu.sg, lijuanzi@tsinghua.edu.cn
[*]Equal contribution.  [†]Co-corresponding authors.

## Abstract

Ultra-long generation by large language models (LLMs) is a widely demanded scenario, yet it remains a significant challenge due to their maximum generation length limit and overall quality degradation as sequence length increases. Previous approaches, exemplified by LongWriter (Bai et al., 2025), typically involve supervised fine-tuning (SFT) on synthetic long-form outputs. However, this strategy heavily depends on synthetic SFT data, which is difficult and costly to construct, often lacks coherence and consistency, and tends to be overly artificial and structurally monotonous. In this work, we propose an *incentivization*-based approach that, starting entirely from scratch and without relying on any annotated or synthetic data, leverages reinforcement learning (RL) to foster the emergence of ultra-long, high-quality text generation capabilities in LLMs. We perform RL training starting from a base model, similar to R1-Zero, guiding it to engage in reasoning that facilitates planning and refinement during the writing process. To support this, we employ specialized reward models that steer the LLM towards improved length control, writing quality, and structural formatting. Experimental evaluations show that our *LongWriter-Zero* model, trained from Qwen2.5-32B, consistently outperforms traditional SFT methods on long-form writing tasks, achieving state-of-the-art results across all metrics on WritingBench and Arena-Write, and even surpassing 100B+ models such as DeepSeek R1 and Qwen3-235B.

## 1 Introduction

Ultra-long text generation that extends over thousands of words has become an increasingly crucial capability for large language model (LLM) in real-world applications such as multi-section report writing, narrative storytelling, legal drafting, and educational content creation (Yao et al., 2019; Schmidgall et al., 2025; Bai et al., 2025; Wu et al., 2025b; Tu et al., 2025a; Wu et al., 2025a). Despite recent advances enabling LLMs to generate outputs exceeding 10,000 words, prior studies have shown that existing long-output LLMs often suffer from issues such as local incoherence, internal contradictions, repetitive phrasing, topic drift, and structural collapse in long-form outputs (Wu et al., 2025c; Bai et al., 2025; Wu et al., 2025b).

Previous efforts have largely relied on extending long-form generation through supervised fine-tuning over synthetic long-output datasets, that is, instruction-output pairs constructed through expert-designed agentic pipelines (Bai et al., 2025; Pham et al., 2024; Tu et al., 2025b; Wu et al., 2025a; Quan et al., 2024). While SFT offers a straightforward approach, it comes with two critical

*Supervised Fine-tuning*

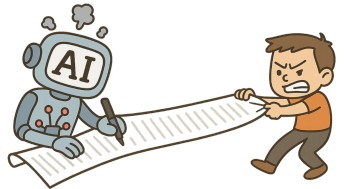

*Reinforcement learning*

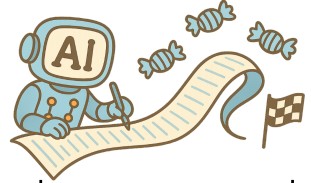

*Ultar-Long text generation*

Figure 1: SFT vs. RL in long-form generation.

limitations. First, the training data is constructed using existing
LLMs, which constrains diversity and innovation in writing styles and inherently caps quality at the level of the off-the-shelf models. Second, the maximum likelihood objective fails to provide explicit signals for optimizing global-level properties such as coherence or formatting consistency (Deng et al., 2022; Pham et al., 2024).

To move beyond these limitations, we pioneer a novel framework for long-form generation: **using reinforcement learning to activate long-form generation abilities in LLMs from scratch**. Rather than fitting on fixed reference texts, RL enables models to optimize for long-range objectives through reward signals that capture desired output qualities, and without the need for manually curated SFT datasets (as illustrated in Figure 1). This mirrors recent advances in reasoning-centric domains such as math and coding, where models like DeepSeek-R1-Zero (DeepSeek-AI et al., 2025a), RL-trained from a base model, have shown notable performance gains. We hypothesize that a similar RL-driven approach can empower LLMs to produce longer, more coherent, and more logically structured outputs aligned with input instructions.

Concretely, we investigate how to effectively train long-form generation policies via RL, addressing three key research questions:

- **RQ1** (**Reward Design**): How can reward models be designed to best guide long-form generation?

- **RQ2** (**Test-time Scaling**): Large reasoning models (LRMs) show substantial gains from inference-time scaling, particularly through long Chain-of-Thought (CoT) (OpenAI, 2024b; DeepSeek-AI et al., 2025a; Yeo et al., 2025). However, prior LRM research focus predominantly in math and code tasks. Can similarly introducing a long CoT enhance RL-based long-form generation?

- **RQ3** (**Impact of Continual Pretraining**): Can continual pretraining on long-form materials and reasoning data further raise the performance ceiling of RL-trained models?

Through systematic ablation and controlled experiments, we find that all three components—reward design, test-time scaling, and continual pretraining—are critical for maximizing the effectiveness of RL in long-form generation. At the same time, we find that RL training significantly outperforms SFT training in long-form writing tasks. Additionally, RL can unlock higher potential in base models with stronger foundational capabilities. We integrate these insights into a final model, *LongWriter-Zero* (trained based on Qwen2.5-32B), which achieves state-of-the-art results on long-form writing benchmarks such as *WritingBench* (Wu et al., 2025d) and *Arena-Write* (Sec. 2.1).

## 2 REINFORCEMENT LEARNING FOR ULTRA-LONG TEXT GENERATION

In this section, we present our reinforcement learning framework designed to unlock ultra-long-form generation capabilities in LLMs. We begin by describing the overall RL setup, including the training algorithm and key components of our framework in Sec. 2.1. We then address the three core research questions outlined in Sec. 1, respectively focusing on: Reward Design (Sec. 2.2), Test-time Scaling (Sec. 2.3), and Impact of Continual Pretraining (Sec. 2.1).

### 2.1 RL SETUP

To train policies for ultra-long-form generation, we adopt the Group Relative Policy Optimization (GRPO) algorithm (Shao et al., 2024; DeepSeek-AI et al., 2025a) for RL training. Below, we detail the core components of our RL setup.

**Training Algorithm.** GRPO extends Proximal Policy Optimization (PPO) (Schulman et al., 2017) by computing normalized advantages over a group of sampled completions. For each training input $q$, it samples a set of candidate outputs $\{o_1, o_2, \cdots, o_G\}$ from the current policy $\pi_{\theta_{old}}$. Each output is scored by a reward model, and the advantage for the $i$th sample scored with $r_i$ is computed as:

$$A_i = \frac{r_i - \text{mean}(\{r_1, \ldots, r_G\})}{\text{std}(\{r_1, \ldots, r_G\})}.$$ (1)

GRPO then maximizes the clipped objective:

$$J_{\text{GRPO}}(\theta) = \mathbb{E}_{q \sim P(Q), \{o_i\}_{i=1}^G \sim \pi_{\theta_{\text{old}}}} \left[ \frac{1}{G} \sum_{i=1}^{G} \min\left( r_i^{\text{ratio}} A_i, \text{clip}(r_i^{\text{ratio}}, 1 - \varepsilon, 1 + \varepsilon) A_i \right) \right.$$

$$\left. - \beta \, D_{\text{KL}}\left( \pi_\theta \,\|\, \pi_{\text{ref}} \right) \right], \tag{2}$$

where the importance sampling ratio $r_i^{\text{ratio}} = \frac{\pi_\theta(o_i|q)}{\pi_{\theta_{\text{old}}}(o_i|q)}$. The hyperparameters $\varepsilon$ and $\beta$ control the clipping threshold and the KL penalty term, respectively. The reference policy $\pi_{\text{ref}}$ is fixed to the initialization of $\pi_\theta$.

**Query Source.** We sample training prompts from two large-scale, real-world instruction datasets: English instructions from *WildChat-1M* (Zhao et al., 2024) and Chinese instructions from *LMSYS-Chat-1M* (Zheng et al., 2023). To ensure the quality and suitability of queries for long-form generation, we use QwQ-32B model (Team, 2025) to filter inputs, retaining only requests that demand high-quality, extended outputs.[1]

**Training Configuration.** We conduct RL training based on the Qwen2.5-32B base model (Team, 2024) using Megatron's reinforcement learning framework on 8 nodes, each with $8 \times$ H800 GPUs. Each optimization step samples 32 concurrent trajectories with a batch size of 32 training prompts. To support long text generation, we set the maximum output length to 14,000 tokens. During training, we adopt sampling with $T = 0.8$ and top-$p = 1.0$. We set $\epsilon = 0.2$ and remove the KL penalty term by setting $\beta = 0$, according to the empirical suggestions in the DAPO algorithm (Yu et al., 2025).

**Evaluation (Arena-Write).** To provide a timely and comprehensive monitor of the model's long-form generation capabilities during training, we manually curate *Arena-Write*, a lightweight benchmark comprising 100 diverse real-world user writing instructions as test prompts, with 40% of the prompts demanding outputs exceeding 2,000 words. Following the automatic evaluation protocol of Arena-Hard (Li et al., 2024), which simulates the crowdsourced Chatbot Arena leaderboard, we conduct pairwise win-rate comparisons for each test prompt against responses from six strong baselines: DeepSeek-R1[2] (DeepSeek-AI et al., 2025a), Qwen2.5-72B (Team, 2024), GPT-4o-2024-11-20 (OpenAI, 2024a), GLM-Z1-32B-0414 (GLM et al., 2024), Claude-3.5-Sonnet (Anthropic, 2024), and DeepSeek-R1-Distill-Qwen-32B (DeepSeek-AI et al., 2025a), yielding a total of 600 comparisons. We use Qwen2.5-72B as the automatic judge to label each comparison as "win", "tie", or "lose" compared to the baseline responses, with the judging prompt provided in Appendix A.5. Evaluation results are reported using Elo ratings (higher means better), computed based on win rates against the baseline responses.

## 2.2 RQ1: Reward Design

In domains such as mathematical reasoning and code generation, reinforcement learning often relies on deterministic rule-based reward by comparing to the groundtruth answer (DeepSeek-AI et al., 2025a; Team, 2025). However, such approaches are generally infeasible for open-ended text generation due to the inherent complexity, subjectivity, and multidimensional nature of evaluating long-form outputs. To address these challenges, we design a composite reward function that integrates multiple reward models, each targeting a distinct aspect of writing quality.

**Length RM.** To guide models toward producing responses of appropriate length—particularly for long-form writing task where outputs may be expected to span thousands of words and responses short of that are typically considered failure to meet task requirements—we design a length reward model that evaluates whether a generated output is too short, overly verbose, or well-aligned with the expected length. Specifically, we employ QwQ-32B (Team, 2025) to predict the appropriate word count range for each query (details provided in Appendix A.3), which serves as the supervisory signal. For example, a query requiring a 3,000-word essay would correspond to a target range of

---

[1]Filtering method details are provided in Appendix A.2.

[2]Throughout our paper, DeepSeek-R1 refers to the version released on 2025-03-27.

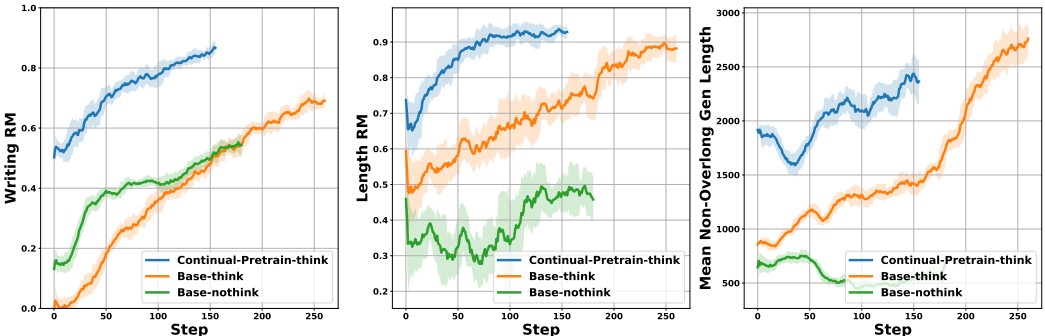

Figure 2: RL Training curves of three setups (*Base-nothink*, *Base-think*, and *Continual-Pretrain-think*) across three metrics: Writing RM (left), Length RM (middle), and Mean Non-Overlong Generation Length (right).

$[2,700, 3,300]$ words. During inference, outputs falling within this length range, i.e., $[L_{\text{lower}}, L_{\text{upper}}]$, receive higher rewards, while under- or over-length responses are penalized:

$$
r_{\text{length}}(o) = \begin{cases} 1, & \text{if } L_{\text{lower}} \leq len(o) \leq L_{\text{upper}}, \\ \dfrac{len(o)}{L_{\text{lower}}}, & \text{if } len(o) < L_{\text{lower}}, \\ \dfrac{L_{\text{max}} - len(o)}{L_{\text{max}} - L_{\text{upper}}}, & \text{if } len(o) > L_{\text{upper}}. \end{cases} \tag{3}
$$

**Writing RM.** The writing reward model $r_{\text{write}}$ is trained on manually-labeled preference data $(y_w, y_l)$ over writing-related prompt $x$. It uses Qwen2.5-72B (Team, 2024) as the backbone and is optimized under the loss function (Rafailov et al., 2024; Hou et al., 2024), following the Bradley–Terry model (Bradley & Terry, 1952) to model preferences:

$$
\mathcal{L} = -\mathbb{E}_{(x, y_w, y_l) \sim D}[\log(\sigma(r_{\text{write}}(x, y_w) - r_{\text{write}}(x, y_l)))]. \tag{4}
$$

The Writing RM aims to capture holistic writing quality, including fluency, coherence, and helpfulness[3]. Through this training process, it learns to provide a high-level reward signal that aligns model outputs with human expectations, encouraging natural, well-structured, and goal-relevant responses.

**Format RM.** To enforce structural integrity and reduce redundancy, the Format RM checks whether the output conforms to a predefined structure: exactly one `<think>` segment followed by one `<answer>` segment. It also penalizes repetitive content based on semantic overlap metrics, which is especially important in RL settings where models may easily reach the target length required by the Length RM by generating nearly identical paragraphs.

**Final Reward.** A naive reward-averaging strategy may cause the overall reward to be dominated by a sub-reward (e.g., length or format) with a larger numeric scale. To prevent this imbalance, each reward component is first *individually normalized within its group* into the same $[-1, 1]$ range before computing its advantage. We then compute the final reward signal through advantage-level averaging:

$$
A_{\text{final}} = \frac{1}{3} \left( A_{\text{length}} + A_{\text{write}} + A_{\text{format}} \right), \tag{5}
$$

which is plugged into the final GRPO objective in Eq. 2. This normalization-then-aggregation scheme ensures that all reward components contribute on a comparable scale, preventing length or format from overwhelming writing quality and promoting a more stable and balanced learning signal for long-form generation.

---

[3] While effective for guiding long-form writing, the RM does not attempt to cover all quality dimensions—particularly fine-grained factuality—which we explicitly acknowledge as a limitation (Appendix A.7). It is designed as a practical, task-aware proxy for long-form quality rather than a fully comprehensive reward.

**Result.** Using the RL setup described in Sec. 2.1, we train the model with the final composite reward. This training setting is denoted as *Base-nothink*. As shown by the green curve in Figure 2, both the Writing RM and Length RM scores improve steadily over RL steps, with particularly pronounced gains in writing quality. Furthermore, performance on Arena-Write improves continuously, with Elo scores increasing from 200 to over 600 (Figure 3 Green). These results demonstrate the effectiveness of our reward design in long-form generation.

## 2.3 RQ2: TEST-TIME SCALING

Recent advances in mathematical and programmatic reasoning, such as DeepSeek-R1 (DeepSeek-AI et al., 2025a) and OpenAI o1 (OpenAI, 2024b), have popularized a new scaling law dimension via test-time scaling: prompting the model to "think" in a dedicated intermediate step before answering through a long Chain-of-Thought (Wei et al., 2022). This *think step*, typically wrapped in `<think>` and `</think>` tokens, allows the model to plan and reflect before committing to a final response. These methods have achieved strong empirical results in reasoning-heavy domains like math and code. However, it remains unclear whether a similar test-time scaling effect generalizes to open-ended tasks such as long-form writing. Writing is inherently subjective and multidimensional, requiring not only coherence and clarity but also tone, structure, creativity, and reader alignment. Whether these qualities benefit from explicit intermediate reasoning optimized through reinforcement learning is an open question.

To investigate this, we compare two prompting strategies during RL training and inference, one explicitly demand the model to engage in thinking and the other asks the model to directly output the response:

> **_Think_ Prompt**
>
> A conversation between the user and the assistant. The user provides a writing/general task, and the assistant completes it. The assistant first deeply thinks through the writing/answering process in their mind before providing the final written work to the user. The assistant should engage in comprehensive and in-depth planning to ensure that every aspect of the writing/general task is detailed and well-structured. If there is any uncertainty or ambiguity in the writing request, the assistants should reflect, ask themselves clarifying questions, and explore multiple writing approaches to ensure the final output meets the highest quality standards. Since writing is both a creative and structured task, the assistant should analyze it from multiple perspectives, considering coherence, clarity, style, tone, audience, purpose, etc. Additionally, the assistant should review and refine the work to enhance its expressiveness. The writing thought process and the final written work should be enclosed within `<think>` and `<answer>` tags, respectively, as shown below:
> `<think>` A comprehensive strategy for writing that encompasses detailed planning and structural design—including brainstorming, outlining, style selection, audience adaptation, self-reflection, quality assurance, etc `</think>`
> `<answer>` The final written work after thorough optimization and refinement `</answer>`

> **_Direct-Answer (Base-nothink)_ Prompt**
>
> A conversation between the user and the assistant. The user provides a writing/general task, and the assistant completes it. The assistant directly provides the final written work. The final written work should be enclosed within `<answer>` tags, as shown below: `<answer>`Final written work.`</answer>`

**Result.** As shown by the yellow curve in Figure 2, models trained with Think Prompt (*Base-think*) initially exhibit lower Writing RM scores—close to zero in early RL steps—compared to their Direct-Answer counterparts (*Base-nothink*). This early lag is expected, as the model must first learn the structure and utility of producing `<think>` and `<answer>` segments driven by the Format RM. However, as RL progresses, the *Base-think* model steadily improves and ultimately surpasses the *Base-nothink* baseline, achieving a higher ceiling in Writing RM scores.

We attribute this improvement to the reflective planning enabled by the `<think>` phase, which helps the model organize thoughts, segment content meaningfully, and allocate information across the output more effectively. Indeed, the model learns to expand this planning phase over time, with the 'think' token length steadily increasing throughout RL training, a dynamic detailed in Appendix

3.5. This is further supported by consistently higher Length RM scores, indicating better control over output length through planning in long CoT. Moreover, as illustrated in Figure 3, *Base-think* significantly outperforms *Base-nothink* on the Arena-Write benchmark (1,200 vs. 700). Together, these results highlight the value of incorporating an explicit *think step* in RL for long-form writing, improving both the internal structure and overall output performance of ultra-long responses.

### 2.4 RQ3: IMPACT OF CONTINUAL PRETRAINING

Prior work has emphasized that the performance ceiling of RL is often bounded by the capabilities of the underlying base model (Yue et al., 2025; Li et al., 2025a; Yeo et al., 2025). In this study, we wonder whether this observation also holds for open-ended tasks like long-form writing, which demand skills such as stylistic control, narrative planning, and length control. To investigate this, we continual pretrain the Qwen2.5-32B model on 30 billion tokens of high-quality, writing-centric data before RL training. The corpus comprises a diverse selection of Chinese and English books, reports, and academic papers, covering a broad spectrum of genres and topics to strengthen the model's writing competence. Additionally, we distill a small portion of long CoT data from the RL trained *Base-think* model (from Sec. 2.3) to enhance the model's reflective reasoning ability and facilitate initial format alignment. This CoT data is incorporated into the pretraining corpus at a minimal ratio of 1% to avoid memorization of

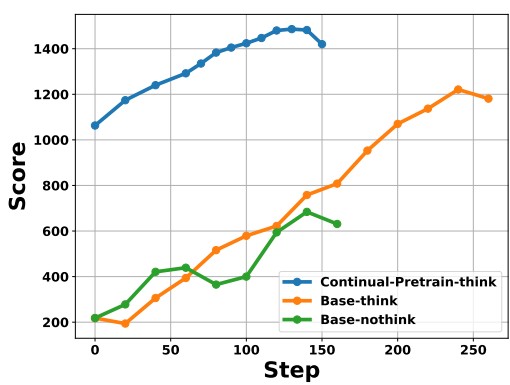

Figure 3: Elo scores evaluated on Arena-Write during training for the three setups: *Base-nothink*, *Base-think*, and *Continual-Pretrain-think*. The y-axis shows the Elo score, and the x-axis represents training steps.

specific CoT patterns. The model is trained with a batch size of 512, using packed sequences with a maximum context length of 32K tokens. A detailed breakdown of the pretraining data is provided in Appendix A.4. All data, except for Ours_think, are sourced from Common Crawl. We apply the same GRPO training to the model after continual pretraining, deriving the *Continual-Pretrain-think* model.

**Result.** As shown by the blue curve in Figure 2, the *Continual-Pretrain-think* model shows higher initial scores in both the Writing RM and Length RM metrics in the beginning. This early advantage roots in improved writing priors acquired during continual pretraining, as well as early format alignment fostered by the inclusion of distilled long CoT data. Beyond a strong starting point, the model also achieves a higher overall reward ceiling. These gains are further reflected in the Arena-Write benchmark, where the model starts with an Elo score of over 1000 and eventually reaches around 1400 at convergence, outperforming the *Base-think* model that does not involve a continual pretraining stage. This performance corresponds to nearly an 80% win rate against strong large reasoning models such as DeepSeek-R1, underscoring the significant benefits of continual pretraining in pushing the RL performance frontier for long-form generation tasks.

## 3 EVALUATION

### 3.1 EVALUATION SETUP

**LongWriter-Zero.** Based on the insights from the three research questions, we establish a robust training pipeline for long-form generation. We begin with continual pretraining on long books and articles to enhance the model's general writing competence. Following this, we apply GRPO training with the *Think* prompt to explicitly encourage the model to engage in reasoning before generating responses. During this stage, the model is optimized with three reward models to better align with the multidimensional long-form writing preferences. Based on this training pipeline, we build our final model, *LongWriter-Zero*, by starting from Qwen2.5-32B and applying 30B tokens of continual pretraining followed by 150 steps of RL. During the evaluation, we use the same prompt format as in

| Models | Avg | Languages | | Domains | | | | | | Requirements | | | | | | Elo |
|---|---|---|---|---|---|---|---|---|---|---|---|---|---|---|---|---|
| | | ZH | EN | D1 | D2 | D3 | D4 | D5 | D6 | R1 | C | R2 | C | R3 | C | |
| *Proprietary LLMs* | | | | | | | | | | | | | | | | |
| GPT-4o-2024-11-20 OpenAI (2024a) | 8.16 | 8.3 | 8.1 | 8.1 | 8.1 | 8.2 | 8.1 | 8.4 | 8.1 | 8.3 | 8.7 | 8.2 | 8.9 | 8.2 | 8.3 | 947 |
| o1-Preview OpenAI (2024b) | 8.15 | 8.1 | 8.2 | 8.0 | 8.1 | 8.2 | 8.2 | 8.4 | 8.1 | 8.2 | 8.6 | 8.2 | 8.8 | 8.2 | 8.2 | 1080 |
| Claude-Sonnet-4 Anthropic (2025) | 8.60 | 8.6 | 8.5 | 8.6 | 8.6 | 8.5 | 8.6 | 8.7 | 8.5 | 8.6 | 8.8 | 8.6 | 9.0 | 8.6 | 8.6 | 1185 |
| Qwen2.5-Max Team (2024) | 8.37 | 8.4 | 8.3 | 8.3 | 8.3 | 8.4 | 8.4 | 8.5 | 8.4 | 8.5 | 8.7 | 8.4 | 9.0 | 8.4 | 8.5 | 1029 |
| *Open-source LLMs* | | | | | | | | | | | | | | | | |
| DeepSeek-R1 DeepSeek-AI et al. (2025a) | 8.55 | 8.7 | 8.5 | 8.5 | 8.5 | 8.6 | 8.6 | 8.7 | 8.6 | 8.7 | 8.9 | 8.6 | 9.0 | 8.6 | 8.7 | 1343 |
| DeepSeek-V3 DeepSeek-AI et al. (2025b) | 7.95 | 8.0 | 7.9 | 7.9 | 7.8 | 8.0 | 7.8 | 8.2 | 8.0 | 8.1 | 8.6 | 8.0 | 8.9 | 8.0 | 8.2 | 1236 |
| Mistral-Large-Instruct Jiang et al. (2023) | 7.64 | 7.6 | 7.7 | 7.7 | 7.6 | 7.8 | 7.3 | 7.9 | 7.6 | 7.7 | 8.2 | 7.7 | 8.7 | 7.7 | 7.9 | 724 |
| Qwen3-235B-A22B Yang et al. (2025) | 8.68 | 8.7 | **8.6** | 8.6 | 8.6 | 8.6 | **8.7** | 8.8 | 8.6 | 8.7 | 8.9 | 8.7 | 9.0 | 8.7 | **8.8** | 1343 |
| Qwen-2.5-72B-Instruct Team (2024) | 7.90 | 8.0 | 7.9 | 8.0 | 7.8 | 8.1 | 7.7 | 8.2 | 7.8 | 8.0 | 8.3 | 8.0 | 8.8 | 7.9 | 8.0 | 911 |
| Qwen-2.5-7B-Instruct Team (2024) | 7.43 | 7.3 | 7.5 | 7.7 | 7.4 | 7.6 | 6.9 | 7.8 | 7.3 | 7.5 | 7.9 | 7.6 | 8.6 | 7.4 | 7.5 | 661 |
| Llama-3.3-70B-Instruct Dubey et al. (2024) | 7.01 | 6.7 | 7.3 | 7.0 | 6.9 | 7.0 | 6.8 | 7.3 | 7.3 | 7.1 | 7.8 | 7.1 | 8.2 | 7.0 | 7.2 | 570 |
| Llama-3.1-8B-Instruct Dubey et al. (2024) | 6.35 | 5.7 | 6.9 | 6.6 | 6.4 | 6.1 | 6.0 | 6.7 | 6.6 | 6.4 | 7.0 | 6.4 | 7.6 | 6.3 | 6.4 | 445 |
| *Capability-enhanced LLMs* | | | | | | | | | | | | | | | | |
| Suri-I-ORPO (7B) Pham et al. (2024) | 4.97 | 4.4 | 5.5 | 5.6 | 5.3 | 5.0 | 4.1 | 5.0 | 5.1 | 4.8 | 5.2 | 5.0 | 5.4 | 4.5 | 4.0 | - |
| LongWriter-8B Bai et al. (2025) | 7.91 | 7.9 | 7.9 | 8.0 | 8.1 | 8.1 | 7.7 | 8.1 | 7.6 | 7.9 | 8.2 | 8.1 | 8.8 | 7.7 | 7.7 | 457 |
| **LongWriter-Zero (32B)** | **8.69** | **8.8** | **8.6** | **8.7** | **8.8** | **8.8** | 8.4 | **8.9** | **8.6** | **8.7** | **8.9** | **8.7** | **9.0** | **8.6** | 8.5 | **1447** |
| *w/o Continual Pretrain* | 8.12 | 8.3 | 8.0 | 8.2 | 8.2 | 8.2 | 7.8 | 8.4 | 8.1 | 8.2 | 8.5 | 8.2 | 8.8 | 7.9 | 7.4 | 1221 |
| *w/o Thinking* | 8.04 | 8.2 | 7.9 | 8.2 | 8.1 | 8.1 | 7.6 | 8.4 | 8.0 | 8.0 | 8.3 | 8.1 | 8.6 | 7.9 | 7.0 | 668 |

Table 1: WritingBench performance of different LLMs across six domains and three writing requirements (scale: 1–10). The domains are: (D1) Academic & Engineering, (D2) Finance & Business, (D3) Politics & Law, (D4) Literature & Art, (D5) Education, and (D6) Advertising & Marketing. Requirements include (R1) Style, (R2) Format, and (R3) Length. "C" denotes the category-specific score. Arena-Write Elo scores are shown in the final red box.

training (*Think* or *No-Think*), as described in Sec. 2.3. For model with long CoT, we evaluate only the final response wrapped between `<answer>` and `</answer>`.

**Benchmark Setup.** We evaluate model performance on three evaluation suites:

**(1) WritingBench** (Wu et al., 2025d) is a comprehensive benchmark for long-form writing, covering six major domains and 100 sub-domains across creative, persuasive, informational, and technical genres. It includes 1,200 real-world writing prompts, each paired with five query-specific evaluation criteria. WritingBench adopts a query-dependent evaluation framework and uses a Qwen2.5-7B critic model fine-tuned on 50K human-annotated samples, achieving 83% agreement with human judgments across dimensions like style, format, and length.

**(2) Arena-Write**, introduced in Sec. 2.1, has 100 instructions; outputs are judged against six baselines by pairwise win rates and summarized with Elo ratings.

**(3) Human-in-the-loop Win-rate Evaluation**. We compile a set of 200 real-world user instructions and compare the win-rate of *LongWriter-Zero* against six leading models, including Qwen3-235B-A22B (Yang et al., 2025), DeepSeek-V3 (DeepSeek-AI et al., 2025b), DeepSeek-R1 (DeepSeek-AI et al., 2025a), Llama-4-Scout (AI, 2025), Claude-Sonnet-4 (Anthropic, 2025) and Gemini-2.5-Pro-0506 (DeepMind, 2025). The win-rate is initially assessed using GPT-4.1 (OpenAI, 2025) with the prompt in Appendix A.5. To further validate these automatic judgments, we also incorporate human evaluations.

## 3.2 MAIN RESULT

We evaluate *LongWriter-Zero* on WritingBench and Arena-Write after continual pretraining and RL. As shown in Table 1, *LongWriter-Zero* achieves the **highest overall critic score of 8.69**, outperforming all models (e.g., Qwen-Max: 8.37, GPT-4o-2024-11-20: 8.16), including the strongest open-source baseline DeepSeek-R1 (8.55). Across different domains, *LongWriter-Zero* obtains the best performance in five out of six domains—Academic & Engineering (8.7), Finance & Business (8.8), Politics & Law (8.8), Education (8.9), and Advertising & Marketing (8.6, tie)—while slightly lagging behind DeepSeek-R1 (8.6) in Literature & Art (8.4). On writing requirement metrics, it also achieves the highest scores in Style (R1: 8.7, C: 8.9) and Format (R2: 8.7, C: 9.0), while maintaining a competitive Length score (R3: 8.6). Meanwhile, *LongWriter-Zero* also significantly outperforms other models in Arena-Write, achieving an Elo rating of 1447, followed by DeepSeek-R1 and Qwen3-235B-A22B, which are tied for second place with a score of 1343.

We also perform an ablation study on our two key strategies: Test-time Scaling and Continual Pretraining. The "-*Continual Pretrain*" model refers to *Base-think* in Sec. 2.3, and further "-*Thinking*" corresponds to *Base-nothink* in Sec. 2.2. We observe that progressively removing these two techniques leads to a substantial performance drop on both WritingBench and Arena-Write. Among them, *Thinking* proves more critical for Arena-Write (1221 ↘ 668), while *Continual Pretraining* plays a more important role on WritingBench (8.69 ↘ 8.12). These ablation results confirm that integrating intermediate reasoning (*think*) and continual pretraining are crucial to *LongWriter-Zero*'s performance, making it highly effective for long-form generation tasks.

## 3.3    SFT VS. RL

In this subsection, we compare the effectiveness of SFT and RL using the same base models: Qwen2.5-32B and our continual trained Qwen2.5-32B in Sec. 2.4. For SFT, we utilize writing instruction data from ShareGPT combined with long-output samples from the LongWriter-6K dataset. Evaluations conducted on Arena-Write are presented in Figure 4. We observe that, over both base models, RL consistently outperforms SFT. The performance of SFT is constrained by the overall quality of its training data, whereas RL continuously improves long-form generation through reward signals. Notably, despite stronger initialization from continual pretraining, the SFT models show only marginal improvement, with scores increasing slightly from 964 (base) to 971 (continual pretrained). In contrast, the RL-based approach significantly benefits from the stronger, continually pretrained initialization, yielding substantial performance gains (1221 ↗ 1447) and clearly surpassing both SFT variants. This suggests that stronger base models can achieve greater performance improvements through RL, whereas the effectiveness of SFT remains constrained by the supervision data.

## 3.4    WIN-RATE RESULT

While WritingBench provides a comprehensive absolute-quality evaluation, relying on a single critic model may introduce model-family biases or score compression among strong systems. To obtain a more robust and model-agnostic view, we additionally adopt a *human-in-the-loop win-rate evaluation*. During evaluation, each query is evaluated twice with swapped response orders to mitigate positional bias, and results are categorized as win, loss, or tie. This pairwise win-rate metric complements WritingBench by capturing finer relative preferences between closely matched models. The win-rate results are shown in Figure 5.

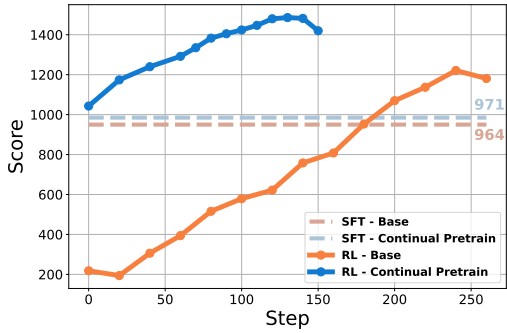

Figure 4: Arena-Write performance across RL training steps, comparing RL (solid) and SFT (dashed) starting from Base (orange) and Continual Pretrain (blue) initializations.

**LLM Evaluation.**   From the six donuts on the left of Figure 5, where results are automatically judged by GPT-4.1, *LongWriter-Zero* consistently demonstrates a substantial performance lead over six strong baseline models. The win rates for our model in these automatic evaluations reach as high as 98.2% and remain above 62% even against the strongest baselines. Despite having only 32B parameters, its long-form generation capabilities rival those of much larger LLMs.

**Human Evaluation.**   To mitigate potential biases in automatic evaluation, we also conduct a supplementary human evaluation, shown in the right two donut charts in Figure 5 (comparing against DeepSeek-R1 and Qwen3-235B-A22B). Three independent annotators with undergraduate degrees evaluated each query according to the same "win/loss/tie" criteria as the automatic evaluation. While the annotators tended to assign ties in cases of subtle differences between responses—slightly lowering the overall win rate—*LongWriter-Zero* still consistently demonstrates strong human preference, validating its reliability in real-world scenarios.

At the end of our experiment, we provide case studies showcasing *LongWriter-Zero*'s long CoT and final responses in Appendix A.10.

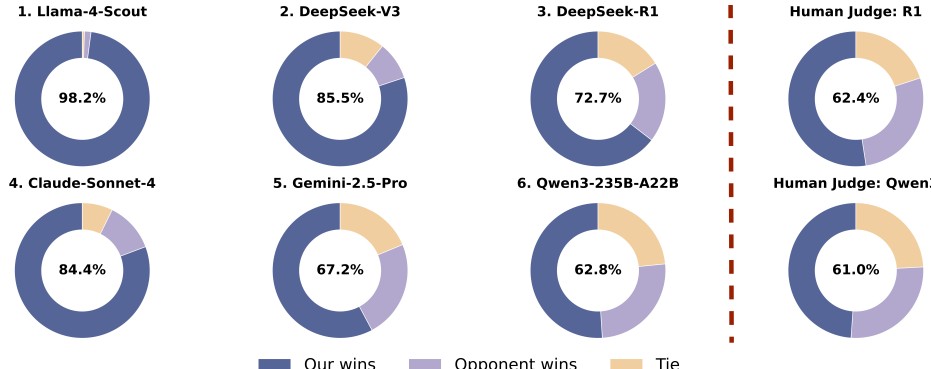

Figure 5: Win-rate results of *LongWriter-Zero* in human-in-the-loop win-rate evaluation. Left six charts: Outcomes judged by GPT-4.1 against six baselines (Llama-4-Scout, DeepSeek-V3, DeepSeek-R1, Claude-Sonnet-4, Gemini-2.5-Pro, Qwen3-235B-A22B). Right two charts: Outcomes judged by human annotators (comparing against DeepSeek-R1 and Qwen3-235B-A22B). The percentage in the center indicates the overall win rate, with ties counted as 0.5 wins for each side.

## 3.5 THINK LENGTH DYNAMICS DURING RL TRAINING

Figure 6 tracks the evolution of the mean "thinking" token length for *Base-think* and *Continual-Pretrain-think* during RL. In contrast to math problems—where chain-of-thought can be expanded (DeepSeek-AI et al., 2025a) almost without limit—writing exhibits a natural saturation point: once the "thinking" budget is sufficient to reliably produce high-quality text, additional thinking (e.g., effectively pre-drafting entire paragraphs) yields little marginal gain while consuming valuable context-window capacity, which can ultimately degrade performance. Consequently, both models converge to a task-optimal thinking length and then plateau, rather than increasing indefinitely. Notably, *Continual-Pretrain-think* consistently maintains a slightly longer thinking length than *Base-think* (i.e., a modest lead), but the gap remains small and does not reflect a qualitative difference in behavior.

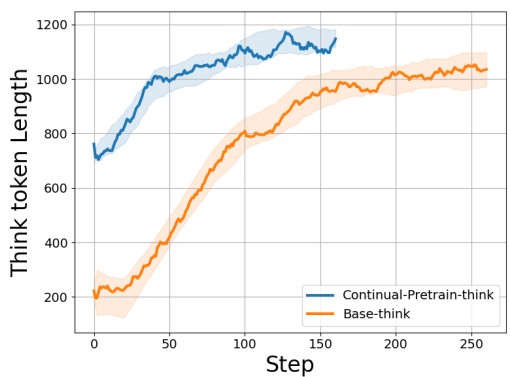

Figure 6: Evolution of the "thinking" token length during RL training. The continually pretrained model consistently generates a longer reasoning chain.

## 4 RELATED WORK

**Long-form Text Generation.** Recent advancements in long-form text generation have explored structured prompting, personalization, and ultra-long output capabilities. Early approaches such as Re3 and DOC (Yang et al., 2022; 2023) introduce recursive or hierarchical strategies to preserve narrative coherence. Later work emphasizes personalization, including LongLaMP (Kumar et al., 2024) and reasoning-enhanced self-training (Salemi et al., 2025), which tailor outputs to user intent. With the increasing demand for ultra-long generation (beyond 2,000 words) (Wu et al., 2025b), new benchmarks and methods have emerged. Suri (Pham et al., 2024) constructs a large-scale instruction-following dataset via back-translation, but its outputs are limited to under 5k tokens and heavily depend on back-translation. LongWriter (Bai et al., 2025) fine-tunes models on agent-generated outputs ranging from 6k to 20k tokens using supervised and preference optimization, achieving desirable ultra-long outputs but inheriting bias from teacher models. Overall, despite progress, current

ultra-long generation techniques are still constrained by synthetic guidance, limiting their ability to ensure coherence, factuality, and generalization over extended contexts.

**RL Scaling for LLMs.**    Recent advancements in LLMs have increasingly leveraged reinforcement learning to enhance reasoning capabilities, particularly in complex tasks such as mathematics, coding, and logical inference. Notable models in this domain include o1 (OpenAI, 2024b), DeepSeek-R1 (DeepSeek-AI et al., 2025a), QwQ-32B (Team, 2025), and Kimi k1.5 (Team et al., 2025), each demonstrating the efficacy of RL in scaling LLM Performance on reasoning tasks. DeepSeek-R1-Zero (DeepSeek-AI et al., 2025a) distinguishes itself by exclusively utilizing RL, foregoing any supervised learning, and relies on GPRO to enhance reasoning without the need for labeled data. Despite having fewer parameters, QwQ-32B (Team, 2025) achieves performance comparable to larger models like DeepSeek-R1 by integrating RL strategies that enhance the model's ability to reason through complex tasks. Kimi k1.5 (Team et al., 2025) is a multi-modal LLM trained with RL, emphasizing long-context understanding and improved policy optimization methods. While these models have successfully scaled RL to enhance reasoning, they primarily focus on short-form tasks and often combine RL with rule-based reward functions. There remains a gap in exploring the exclusive use of RL for scaling LLMs in long-form generation tasks. To address this gap, we propose a novel framework that employs RL exclusively to scale LLMs for long-form generation tasks, and successfully *LongWriter-Zero* with it. By eliminating reliance on supervised learning, *LongWriter-Zero* demonstrates the potential of pure RL approaches in enhancing the coherence, relevance, and overall quality of long-form generation tasks.

## 5    CONCLUSION

This work presents the first attempt to apply RL to ultra-long text generation without relying on synthetic or annotated datasets. By leveraging composite reward models targeting length control, writing quality, and formatting consistency, our method addresses core challenges in long-form generation such as length following, coherence degradation, and structural drift. Experiments on WritingBench, Arena-Write, and human evaluations show that *LongWriter-Zero*, trained with our approach, significantly outperforms both SFT baselines and leading reasoning-based models. Three key insights emerge: (1) tailored reward design is essential for guiding long-form generation; (2) incorporating explicit reasoning steps via the *Think* Prompt during RL enhances planning and coherence; and (3) continual pretraining substantially raises RL performance ceilings. In summary, *LongWriter-Zero* establishes a strong RL-only paradigm for long-form generation, offering new pathways for scalable and coherent ultra-long text production.

## ACKNOWLEDGEMENT

The research is supported by the National Research Foundation, Singapore under its National Large Language Models Funding Initiative (AISG Award No: AISG-NMLP-2024-005), and by AI Singapore under its AI Governance Research Grant (AISG3-GV-2023-010). This work is also supported by the National Natural Science Foundation of China (Grant No. 62476150) and the Beijing Natural Science Foundation (Grant No. L243006). Any opinions, findings, and conclusions or recommendations expressed in this material are those of the author(s) and do not reflect the views of the National Research Foundation and AI Singapore.

## ETHICS STATEMENT

We are committed to upholding the principles outlined in the ICLR Code of Ethics. Our research incorporates a dataset annotated by three professional human annotators. All annotators involved in this project were provided with clear guidelines and gave their informed consent prior to participation. We ensured they were compensated fairly for their labor and time, in line with ethical standards for human-participant research.

The data presented to the annotators was carefully processed to remove any personally identifiable information (PII), thereby protecting the privacy of both the annotators and any individuals potentially represented in the data. Furthermore, we have conducted a thorough assessment of the potential broader impacts of our work. This includes evaluating risks related to the amplification of social biases, the generation of toxic or harmful content, and dual-use applications. Our analysis indicates that our methodology and findings do not introduce or exacerbate these potential harms. The authors declare no competing interests.

## REPRODUCIBILITY STATEMENT

We are committed to ensuring the reproducibility of our work. All datasets generated and used for continual pretraining, reinforcement learning, and evaluation are detailed in the Appendix and will be provided in the supplementary materials. Upon publication, we plan to release the source code for our training framework, the reward models, and the final *LongWriter-Zero* model weights to the public. Detailed descriptions of our experimental setup and hyperparameters are provided throughout the paper and in the Appendix to allow for the full replication of our results.

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

# A  APPENDIX

## A.1  USE OF LARGE LANGUAGE MODELS (LLMS)

Large Language Models (LLMs) were used exclusively for language polishing and improving the clarity of the manuscript. They did **not** contribute to research ideation, methodological design, experimental execution, data analysis, or any other substantive aspects of this work. All scientific content, results, and conclusions are the sole responsibility of the authors.

## A.2  WRITING-TASK SELECTION AND LENGTH-RANGE PREDICTION WITH QWQ-32B

We reformulate the pipeline to *(i)* decide whether a query requests **original writing**, and, if so, *(ii)* predict a suitable `[min, max]` word-count range in one shot. The few-shot prompt below ("Prompt-WL") is fed to the QwQ-32B model.

---

**Writing-Task Selection**

**Task Goal**
Given a user query,
1. Decide if it asks for *original written content*.

---

2. If **NotWriting**, stop.

3. If **Writing**, output a reasonable word-count range "[lower, upper]" (ignore ±10%).

**Response Format**

• If not writing – respond exactly: **NotWriting**.

• If writing – respond with **only** the code block {"range": [lower, upper]}

**Heuristics for Range Estimation**

1. **Depth & Complexity**: more analysis → higher upper bound.

2. **Scope**: multiple sub-topics/sections → longer.

3. **Requested Form**: tweets/notes (0–300); short blog/letter (300–800); school essay (800–1 200); report/article (1200–2500); thesis/proposal/business plan (4000–10000).

4. **Explicit Length Clues**: honour any word/page requirement if stated.

**Few-Shot Examples**

**Example 1**

Query: Write a Weibo post titled "Tips for Preparing for College Final Exams."

Answer: {"range": [0, 300]}

**Example 2**

Query: Translate "Seize the day" into Spanish.

Answer: NotWriting

**Example 3**

Query: Draft a comprehensive 10-page business plan for a new cat-litter product.

Answer: {"range": [4000, 6000]}

Query: **User Query**

Answer:

## A.3  LENGTH RM PROMPT

---

**Appendix: Query Length Assessment Prompt**

You are a professional query text–length assessor. Based on the type of the query content, you should:

1. Deeply understand the core requirement of the query (e.g., essay, blog post, summary, outline, thesis section, etc.). For example, the query "How do I start writing my thesis from scratch" asks for guidance on "how to begin writing a thesis," so you would estimate a word-count range of [400, 800], rather than the total words needed to complete the entire thesis [7000, 10000].

2. Choose a lower bound that is a multiple of 100, with a minimum of 0.

3. Choose an upper bound that is a multiple of 100, with a maximum of 12,000. If the reasonable range certainly exceeds these limits, output: {"range": [0, 0]}

4. Ignore the 10% of extreme length cases to keep the range reasonable for most scenarios, and ensure the difference between upper and lower bounds does not exceed 3,000.

5. If the query contains an explicit word-count requirement, set the range to ±10% of that number. - For "write a 2,000-word essay," output: {"range": [1800, 2200]} - For "no more than 2,000 words," output [1800, 2000]; for "at least 2,000 words," output [2000, 2200].

6. If the query cannot be fulfilled under the given conditions—for example, "Read and analyze this paper" without providing the paper, or "Analyze a project's prospects" without specifying the project details—then output: {"range": [0, 0]}

Example:

Input "Write a high school essay" → {"range": [800, 1000]}

Input "Complete an academic paper on green cities" → {"range": [6000, 10000]}

Please process the current query: **User Query**

Analyze and output the JSON range accordingly.

---

## A.4  CONTINUAL PRETRAIN DATA

The corpora listed in Table 2 were used for continual pre-training. All sources were de-duplicated, language-filtered, and truncated to a maximum sequence length of 32,768 tokens before up-sampling.

Table 2: Continual pretraining data distribution.

| Data | Percentage (%) | Description |
|------|----------------|-------------|
| Long_zh_fiction | 40% | Chinese novels |
| Long_en | 30% | English fiction and non-fiction |
| Long_zh_nonfiction | 15% | Chinese non-fiction books |
| Online_information | 8% | Web novels, posts, and blog articles |
| Long_finance | 5% | Industry reports |
| Long_essay | 1% | Academic papers |
| Ours_think | 1% | Long CoT samples generated by *Base-think* model |

## A.5 EVALUATION PROMPT FOR WIN-RATE JUDGMENT

WritingBench adopts a critic model to evaluate model outputs by assigning scores (ranging from 1 to 10) across 4-5 distinct dimensions. However, this evaluation approach has several limitations. First, due to the relatively small size of the critic model, it may be vulnerable to some word/sentence hacking. Second, we observe that WritingBench primarily focuses on formal or professional writing tasks—such as summaries and reports—whereas our analysis of real-world data from *WildChat* (Zhao et al., 2024) and *LMSYS-Chat-1M* (Zheng et al., 2023) reveals that creative writing (e.g., storytelling, fiction) constitutes a significant portion of user queries.

To address these concerns, we adopt a more direct and interpretable evaluation metric: **win rate**. We evaluate model performance on nearly 200 real-world user queries collected from the aforementioned datasets. For each query, responses are generated by *LongWriter-Zero* and six baseline models. These responses are then compared in a pairwise manner using GPT-4.1-2025-04-14[4] (OpenAI, 2025). To mitigate positional bias, we conduct two evaluations per pair by swapping the response order: `Evaluation_Prompt + A + B` and `Evaluation_Prompt + B + A`. Based on the two judgments, we categorize the results into win, loss, or tie.

---

**SYSTEM_PROMPT for Win-Rate Evaluation in Arena-Write**

Please act as an impartial judge and evaluate the quality of the written responses provided by two AI assistants to the user's writing prompt below. You will be given Assistant A's response and Assistant B's response. Your job is to determine which assistant's writing is superior.
Evaluate them on the following criteria:
**1. Relevance and Completeness**: Does the assistant fully respond to the writing prompt? Does the length meet the user's query expectations? Is the content relevant to the topic, and does it provide sufficient depth, length, and detail, rather than drifting off-topic or simplistic?
**2. Writing Quality**: Evaluate whether the assistant's writing is clear, fluent, and free of obvious grammatical errors. The overall quality of the writing is high, with elegant.
**3. Creativity and Originality**: If applicable, assess the creativity of the response. Does the assistant offer fresh perspectives, unique insights, or demonstrate a certain level of originality?
**4. Specificity and Detail**: Determine whether the assistant provides concrete examples or detailed explanations. Properly justified repetition is permissible.
**5. Tone and Style**: Is the tone appropriate for the writing prompt? Is the writing style consistent throughout? Consider whether it aligns with the expectations of the intended audience or writing purpose.
After evaluating each response, determine which one is superior based on the factors above. Provide your explanation and then select one of the following final verdicts:

- Assistant A is significantly better: `[[A»B]]`
- Assistant A is slightly better: `[[A>B]]`
- Tie, relatively the same: `[[A=B]]`
- Assistant B is slightly better: `[[B>A]]`
- Assistant B is significantly better: `[[B»A]]`

---

[4]Due to the issue of preference leakage, where models tend to favor others from the same developer, we ensure that all baseline models come from companies different from that of the judge model (Bai et al., 2024; Li et al., 2025b).

> Example output: `My final verdict is tie: [[A=B]].`

## A.6 LIMITATION

While *LongWriter-Zero* demonstrates strong long-form generation ability, it is *not* immune to reward-model exploitation. Heuristic rewards—particularly those involving length and style—naturally introduce opportunities for reward hacking, and several concrete failure modes were observed during training.

**(1) Stereotypical or templated stylistic openings.** The policy occasionally adopted highly formulaic openings (e.g., "Early morning...") because such patterns reliably yield favorable scores under the Writing RM. These stylistic shortcuts increase perceived coherence without providing genuine narrative substance.

**(2) Pattern-level and near-duplicate repetition.** Despite sentence-overlap and n-gram penalties, the model sometimes inserted slightly altered repeated passages to satisfy length requirements. Such paraphrased or token-level–modified redundancies are harder to detect and represent a common form of reward hacking in long-output RL training.

**(3) Keyword-driven reward inflation.** As with many preference-trained reward models, the Writing RM exhibits mild biases toward certain sophisticated-sounding terms. The policy occasionally exploited this by inserting high-value keywords even when semantically unnecessary, producing text that appears more "expert-like" without corresponding content depth.

These behaviors highlight a broader limitation of model-based RL: policies can exploit superficial correlations in the reward models rather than fully aligning with human intent. In our work, we introduced stronger mitigation mechanisms—such as format-level structural checks and a multi-stage repetition-detection module that captures pattern-level redundancy—which substantially reduce the most common hacks. However, these measures do not eliminate the issue entirely.

Future directions include developing more discourse-aware and factuality-sensitive reward models, incorporating adversarial or uncertainty-aware objectives to resist keyword and repetition exploitation, and maintaining periodic human-in-the-loop auditing to detect emerging reward-hacking strategies. Overall, we view LongWriter-Zero as a step toward understanding how length-aware rewards behave in practice, rather than a complete solution to RM hacking.

## A.7 ADDITIONAL RESULTS ON SMALLER MODEL SCALE: LONGWRITER-ZERO-14B

To examine whether our reinforcement-learning paradigm generalizes across different model scales, we additionally trained a smaller **Qwen2.5-14B** model using the full LongWriter-Zero pipeline (including task-aware length RM, Writing RM, format-level constraints, and repetition-control mechanisms). The results show that the paradigm remains effective even at significantly smaller scale, achieving consistent improvements over the SFT baseline across both WritingBench and ArenaWriter evaluations.

| Model | WritingBench ↑ | ArenaWriter ELO ↑ |
|---|---|---|
| Qwen2.5-14B-Instruct | 7.60 | 744 |
| LongWriter-Zero-14B | **8.49** (+0.89) | **1198** (+454) |

Table 3: **LongWriter-Zero-14B achieves strong gains at smaller scale.** Even at 14B parameters, our RL framework produces substantial improvements in both absolute critic scores (WritingBench) and pairwise preference evaluations (ArenaWriter ELO).

## A.8 ADDITIONAL BASELINE: LONGWRITER-ON-32B SFT

To provide a clearer comparison with synthetic-data SFT pipelines, we additionally include an SFT baseline obtained by training the **Qwen2.5-32B** model using the publicly released synthetic data from the LongWriter paper (Bai et al., 2025). This configuration can be viewed as a 32B-scale version of the "LongWriter" pipeline, and thus serves as a natural "**LongWriter-on-32B**" baseline.

The results below correspond to the experimental setting used in Section 3.3 (SFT vs. RL). As shown in Table 4, the synthetic-data SFT baseline performs noticeably worse than our RL-based LONGWRITER-ZERO-32B, reinforcing the effectiveness of our proposed RL paradigm over standard synthetic-data SFT approaches.

Table 4: **LongWriter-on-32B SFT baseline.** We evaluate Qwen2.5-32B trained with LongWriter synthetic data against our LONGWRITER-ZERO-32B. Our RL pipeline achieves substantial improvements on both WritingBench and ArenaWriter ELO.

| Model | WritingBench ↑ | ArenaWriter ELO ↑ |
|---|---|---|
| Qwen2.5-32B *(LongWriter-SFT)* | 8.08 | 964 |
| LONGWRITER-ZERO-32B | **8.69** (+0.61) | **1447** (+483) |

## A.9 ADDITIONAL EVALUATION ON LONGBENCH-WRITE

For completeness and comparability with LongWriter (Bai et al., 2025), we evaluate LongWriter-Zero-32B on the LongBench-Write benchmark. Results are shown in Table 5. LongWriter-Zero-32B achieves the strongest performance across all three metrics, indicating that the gains observed on WritingBench transfer to the independent LongBench-Write evaluation.

| Model | $\bar{S}$ | $S_l$ | $S_q$ |
|---|---|---|---|
| Claude 3.5 Sonnet | 80.7 | 73.7 | 87.7 |
| GPT-4 Turbo | 67.3 | 47.9 | 86.6 |
| GLM-4-9B-Chat | 68.3 | 51.0 | 85.5 |
| Llama-3.1-8B-Instruct | 60.3 | 50.0 | 70.6 |
| Llama-3.1-70B-Instruct | 65.6 | 50.8 | 80.3 |
| Mistral-Large-Instruct | 77.0 | 65.6 | 88.3 |
| Suri-I-ORPO | 56.6 | 59.6 | 53.5 |
| LongWriter-8B | 79.8 | 77.4 | 82.2 |
| LongWriter-9B | 80.5 | 78.6 | 82.3 |
| LongWriter-9B-DPO | 84.0 | 82.6 | 85.4 |
| **LongWriter-Zero-32B (ours)** | **86.3** | **84.2** | **88.4** |

Table 5: LongBench-Write results. Baseline scores follow Bai et al. (2025). LongWriter-Zero-32B obtains the best overall quality ($\bar{S}$), long-horizon structure ($S_l$), and content quality ($S_q$).

## A.10 CASE STUDY

We present three case studies comparing our model with Gemini-2.5-Pro (DeepMind, 2025), DeepSeek R1 (DeepSeek-AI et al., 2025a), and Qwen3-235B-A22B (Yang et al., 2025).

> **Query**
>
> Write a 400-word extremely scary ghost story in the first person, with clear logic and engaging content.

> **LongWriter-Zero Think**
>
> The user wants a 400-word first-person ghost story that's extremely scary, with clear logic and engaging content. Let me start by breaking down their request. They specified "extremely scary," so I need to focus on creating a visceral, atmospheric sense of terror. First-person perspective is essential for immediacy and personal connection. Clear logic means the plot needs a structured progression, avoiding leaps that might confuse the reader. Engaging content requires strong character development and a compelling narrative arc. First, I'll brainstorm settings. A haunted house is a classic, but maybe something more specific. A childhood home that's been abandoned could work—nostalgia mixed with

fear. The protagonist returning after a long absence adds tension. The ghost should have a backstory that ties into the setting, perhaps a tragic event. Maybe the protagonist's mother died there, and the ghost is a manifestation of her lingering presence.

Next, the structure. The story should start with the protagonist's return, setting the mood. Introduce the ghost gradually, building dread. Include sensory details like creaking floorboards, cold spots, or strange noises. The climax could involve a confrontation, where the ghost's true nature is revealed. The ending should leave the reader unsettled, maybe with an unresolved threat.

Style and tone are crucial. First-person allows for internal monologue, which can convey fear and uncertainty. The language should be descriptive but not overly flowery. Use short, sharp sentences to mimic the protagonist's racing heart. Sensory details are key: sight, sound, touch, smell, taste. For example, the musty smell of the house, the chill that seeps into the skin, the sound of a child's laughter in an empty room.

The audience is likely readers who enjoy horror stories, so the story needs to be relatable but with a twist. Maybe the ghost isn't a traditional specter but something more symbolic. The purpose is to scare, so the ending should linger, perhaps with a final ominous detail that suggests the threat continues.

Possible plot points: The protagonist returns to the house after a decade. They find it in disrepair, triggering memories. They hear a child's laughter, which grows louder. They discover a room with a rocking chair, and the ghost appears as a shadow. The ghost reveals itself to be the protagonist's mother, who died giving birth to them. The mother's spirit is trapped in the house, unable to rest. The ending could be the protagonist fleeing, but the ghost's laughter follows them, suggesting it's never truly gone.

I need to ensure the logic is clear. The protagonist's return is motivated by a family heirloom, adding a reason for the visit. The ghost's presence is tied to the mother's tragic death, giving it a backstory. The climax should show the ghost's desperation, making it more sympathetic yet terrifying. The ending leaves the reader questioning whether the protagonist truly escaped.

Potential pitfalls: Making the ghost too cliché. To avoid this, I'll give it a unique trait, like the rocking chair, which becomes a symbol of its confinement. Also, the protagonist's emotional journey—fear, nostalgia, guilt—can add depth to the story.

Now, structuring the story into four parts: introduction, rising tension, climax, resolution. Each part should be concise but impactful. Use dialogue sparingly, perhaps a haunting whisper or a child's voice. The final 400 words need to be tight, so each sentence must serve a purpose.

I'll also consider the reader's emotional arc. Start with curiosity, build to fear, then to horror, and finally to dread. The ending should leave the reader unsettled, perhaps with a lingering image or sound.

Finally, review for coherence, clarity, and style. Check for any logical inconsistencies and ensure the first-person perspective stays consistent. Use strong verbs and vivid imagery to enhance the horror. Maybe include a metaphor, like the house as a living creature, to deepen the atmosphere.

Now, time to draft the story with these elements in mind, keeping the word count in check and maintaining a steady pace of tension and terror.

## LongWriter-Zero Response

The creak of the front door echoed like a dying man's gasp as I stepped into the house. I'd been gone ten years, but the air still smelled like my mother's perfume—lavender and cigarette smoke. The wallpaper was peeling, the windows cracked, but the rocking chair in the corner of the living room was the same, its wooden slats worn smooth by time. I'd spent my childhood in that chair, my mom's hands on my shoulders, rocking me to sleep while she hummed lullabies.

I was there to retrieve my grandmother's locket, a family heirloom left on the dining table. The house felt like a lung holding its breath. I tiptoed past the kitchen, where the fridge hummed like a giant insect, and into the hallway. That's when I heard it: a child's laughter, high and shrill, coming from the nursery. My heart leaped. The nursery had been sealed off since my mom died there, giving birth to me. The door was locked, but the laughter grew louder, as if the room itself was mocking me.

I fumbled for the key, my hands slick with sweat. The lock clicked open, and I pushed the door. The room was empty, the crib still standing, its bars rusted like old teeth. But the rocking chair in the corner was moving—slowly at first, then faster, the wood creaking in

time with the laughter. I stumbled backward, tripping over my own feet. The chair stopped. Silence fell like a guillotine.

Then the rocking started again, and I saw it: a shadow on the wall, the shape of a woman. My mother. She'd died giving birth to me, but her ghost was here, trapped in the house she'd loved. I could see her face now, her eyes wide and pleading, her mouth moving. "I'm sorry," she whispered, her voice a scratchy echo. "I'm so sorry."

I ran, my footsteps echoing like gunshots. The front door slammed behind me, and I sprinted down the street, my lungs burning. But the laughter followed me, a high-pitched giggle that seemed to come from everywhere. When I looked back, the house was dark, but the rocking chair was still moving in the nursery window, its shadow swaying like a pendulum. I knew then that she'd never leave—that she'd be waiting, rocking, until the day I returned.

### Gemini-2.5-Pro Response

It started with the rocking chair. I bought it at an estate sale, a heavy oak piece with a dark, almost black finish. I put it in the corner of my bedroom. The first night, I woke up to find it in the middle of the room, facing my bed. I blamed a sloping floor and my own exhaustion, pushing it back against the wall.

The next night, it was closer. Close enough that my feet would have touched it if I'd stretched. My blood ran cold. This wasn't a slanted floor. I spent the next day trying to debunk it, checking for drafts, vibrations, anything. Nothing. That night, I decided to watch it.

I sat up in bed, my eyes locked on the chair in the corner, the moonlight painting it in shades of grey. For hours, nothing happened. My eyelids felt like sandpaper. The urge to blink was a physical agony. Finally, I couldn't hold out. I blinked.

SCRAPE.

The sound was violent, a splintering shriek of wood on wood. My eyes flew open. The chair was now a foot from my bed. In the seat, where there was nothing before, sat the deep, dark impression of a human form. I didn't blink again. I couldn't. My heart hammered against my ribs, a frantic prisoner. I stared at the empty space in the chair, the dent in the darkness, and I understood the rule: It only moves when I can't see it.

My eyes burned, tears streaming down my face, blurring my vision. Each time a tear washed over my pupil, the chair scraped closer. It was herding me. I started backing away, my body trembling, my gaze fixed on the chair. My back hit the bedroom door. My hand fumbled behind me for the knob.

The chair was now at the edge of my bed. I could see the shape in it more clearly – the gaunt hollows of a face, a grin that looked like chipped wood. I just had to get the door open. My fingers finally closed around the cold brass knob.

That's when I heard the soft, dry click of the deadbolt sliding into place.

The sound hadn't come from the door in front of me. It came from the hallway, just behind it. The thing in the chair wasn't trying to get to me. It was a distraction. It was just keeping me looking forward, my back to the door, while its companion locked me in with it.

### Query

Hello, you're a senior HR manager at a company specializing in selling women's clothing on Amazon US. The company has been operating on the Amazon US platform for seven years and has established a certain brand presence. Now, you need to recruit a Senior Amazon Operations Manager with relevant experience, particularly in promoting best-selling products. The candidate should have experience managing a clothing category with fewer than 5,000 SKUs and possess strong English skills.

### LongWriter-Zero Think

The user wants me to draft a job posting for a Senior Amazon Operations Manager at a women's clothing company on Amazon US. Let me start by breaking down the key requirements. The company's been on the platform for seven years, so they're established but maybe looking to scale up. They need someone with experience promoting best-sellers, especially in a clothing category with under 5,000 SKUs. Strong English skills are essential,

so perhaps the candidate will be handling international customers or communicating with Amazon's US-based teams.

First, I need to structure the job posting. Typically, these start with a company overview, then the role's purpose, responsibilities, qualifications, and perks. The user mentioned "senior," so the candidate should have leadership experience. Maybe include a bullet point about mentoring or leading a team.

The clothing category is a specific detail. I should highlight that the candidate needs to have managed similar inventory sizes, maybe 3-5k SKUs. The user also specified fewer than 5,000, so I'll make sure to use that exact number. English skills are crucial, so perhaps include a note about effective communication, maybe even proficiency in writing product descriptions or handling customer inquiries.

I need to think about the audience. This is for experienced professionals, so the tone should be professional but also inviting. Use active voice and bullet points for clarity. Maybe include a sentence about the company's growth and the role's impact on that.

Possible challenges: Ensuring the job posting is specific enough without being too restrictive. The user wants someone with Amazon US experience, so I should mention that the candidate should have worked on the US platform. Also, promoting best-selling products—maybe the candidate should have a track record of increasing sales or improving ranking.

Should I include a section on what the company offers? Perks like flexible hours, benefits, or opportunities for growth. This could make the posting more appealing. Maybe mention the company's brand presence and the chance to work on a well-established platform.

I should also think about keywords. Terms like "Amazon US," "Senior Operations Manager," "clothing category," "SKU management," "best-seller promotion," "English proficiency," "leadership experience." Including these will help candidates find the posting and ensure it's relevant to their skills.

Potential ambiguity: The user says "promoting best-selling products." Does that mean creating new best-sellers or managing existing ones? I'll assume both, so mention driving sales growth and improving product performance. Also, "fewer than 5,000 SKUs" is a specific number—make sure to use that exactly.

Style and tone: Keep it formal but not too stuffy. Use bullet points for the responsibilities and qualifications. Maybe start with an engaging opening line about the company's success and the role's importance.

Audience adaptation: This is for someone with 5-7 years of experience in Amazon operations, so the language should be industry-specific but not overly technical. Highlight the need for strategic thinking, data analysis, and problem-solving skills.

Quality assurance: Review the structure to ensure all requirements are covered. Check for clarity in each section. Maybe ask myself: Would a candidate with the right experience find this posting appealing? Does it clearly outline what's expected?

Potential areas to expand: Mentioning the company's growth trajectory, the competitive nature of the fashion industry, and how the role will contribute to the company's goals. Also, the importance of English skills in dealing with customers, suppliers, or Amazon policies.

In conclusion, the job posting needs to balance the company's history and achievements with the specific skills required for the role. Emphasize the candidate's impact on sales, product management, and team leadership. Use clear, concise language and highlight the benefits of working for a successful, growing company on a major e-commerce platform. Make sure the English proficiency requirement is clear and that the candidate understands the role's scope, from data analysis to strategic planning.

---

**LongWriter-Zero Response**

**Senior Amazon Operations Manager, Women's Clothing | Your Company Name | Remote / Your Preferred Location | Competitive Salary + Benefits**

At *Your Company Name* (YCN), we're redefining the way women experience style and confidence through our curated collections on Amazon US. With seven years under our belt, we've built a loyal customer base and established a brand known for quality, inclusivity, and accessibility. Now, we're scaling—and we need a seasoned Amazon operations leader to drive our next chapter. If you're a data-driven, customer-centric strategist who thrives on turning best-sellers into category icons, this role is for you.

**The Role:**

As our Senior Amazon Operations Manager, you'll own the entire product lifecycle for our women's clothing line, ensuring every item meets our high standards of excellence. Your mission: to elevate our best-selling products to new heights while maintaining a 4.8+ rating average and steady market share growth. You'll collaborate cross-functionally with marketing, merchandising, and customer service teams to execute flawless launches, optimize ad spend, and refine pricing strategies. Key responsibilities include:

- **Driving Product Visibility & Sales:** Analyze competitive data to identify high-potential SKUs (targeting our <5,000 items) and develop tailored promotion plans that balance organic growth with paid advertising. Leverage Amazon's tools (e.g., A+ content, Vine Voice, influencer outreach) to amplify customer engagement and increase review velocity.

- **Inventory & Supply Chain Mastery:** Manage SKUs efficiently, balancing stock levels to meet demand while minimizing markdowns. Negotiate with suppliers to secure competitive pricing and ensure on-time deliveries to Amazon FBA warehouses, optimizing freight costs without compromising speed to market.

- **SEO & Pricing Strategy:** Lead SEO audits to optimize product listings for keywords, ratings, and reviews. Implement dynamic pricing models that adjust to sales trends, competitor actions, and Amazon algorithms to maximize profitability without sacrificing competitiveness.

- **Customer Obsession:** Champion a customer-centric approach by monitoring feedback loops, addressing negative reviews proactively, and translating customer insights into actionable merchandising strategies. Partner with our customer service team to resolve escalations and maintain a best-in-class shopping experience.

- **Stakeholder Alignment:** Communicate complex operations data to internal and external stakeholders, translating metrics into stories that drive decision-making. Work closely with our marketing team to align ad spend with product performance, ensuring every dollar invested delivers ROI.

**What We're Looking For:**

This role demands a unique blend of Amazon expertise, fashion category knowledge, and analytical prowess. Here's what we're looking for in our next superstar:

- **7+ years** of Amazon Vendor Central experience, preferably in women's clothing or a comparable fast-moving consumer goods category with <5,000 SKUs.

- **Proven track record** of managing best-seller promotions that doubled or tripled sales within 6–9 months, with clear data to back your strategies.

- **Fluent English** (written and spoken) for seamless communication with customers, vendors, and cross-functional teams. Strong presentation skills are a must.

- **Analytical acumen** to interpret Amazon reporting (e.g., SALESrank, Impressions, Conversion Rate) and translate insights into actionable plans.

- **Adaptability** to navigate Amazon's evolving policies and market dynamics, with the ability to pivot strategies quickly without losing momentum.

- **A passion for fashion** and a genuine interest in building a brand that empowers women through style. Bonus points if you've worked with Amazon private label brands.

**What We Offer:**

Beyond a competitive salary and comprehensive benefits, you'll join a company that values innovation, collaboration, and growth. Here's why you'll love working at YCN:

- **Ownership of Your Impact:** You'll have a direct line to the CEO, enabling you to shape strategy and see your ideas transform our Amazon presence.

- **Growth Mindset:** We're expanding our product portfolio and looking to launch new lines. Your insights will help define our future direction.

- **Innovative Culture:** From sustainable packaging to digital-first marketing, we're always pushing boundaries. We'll support you in developing new tools and processes that elevate the industry standard.

- **Career Trajectory:** This role is a stepping stone to larger leadership opportunities. We invest in our team's growth, offering mentorship, training, and a clear path to advancement.

If you're ready to take your Amazon ops career to the next level and help build a brand that resonates with millions of women, apply **[here]** with your resume, a brief cover letter explaining how your past achievements align with our goals, and a portfolio of your most successful Amazon campaigns. We can't wait to meet you!

*P.S. We're open to remote candidates based in the US. Let's chat!*

## Qwen3-235B-A22B Response

**Job Title: Senior Amazon Operations Manager (Remote - U.S. Based)**

**Company Overview**
For over seven years, our company has been a trusted name in the women's clothing space on Amazon US. With a strong brand presence and a commitment to quality, style, and customer satisfaction, we've built a loyal customer base and a portfolio of top-selling products. As we continue to grow, we seek a strategic and results-driven Senior Amazon Operations Manager to lead our marketplace strategy, optimize product performance, and drive scalable success in the competitive apparel industry.

**Job Summary**
We are seeking a seasoned Amazon Operations Manager with deep expertise in managing clothing categories (under 5,000 SKUs) and a proven track record of scaling products to bestseller status. This role requires a hands-on leader who can oversee end-to-end Amazon operations, from listing optimization and inventory management to data-driven strategy development. The ideal candidate will have strong English communication skills, a passion for e-commerce, and a demonstrated ability to boost visibility and sales for women's apparel brands.

**Key Responsibilities**

- **Operations Leadership:** Manage daily Amazon US operations, including listing creation/optimization, pricing strategy, and adherence to Amazon's policies and best practices.
- **Product Promotion:** Develop and execute strategies to promote top-selling products, leveraging Amazon Ads (Sponsored Products, Brands, and Display), A+ content, and promotional campaigns.
- **Inventory Management:** Monitor inventory levels, forecast demand, and collaborate with suppliers to ensure optimal stock availability and reduce FBA costs.
- **Data Analysis:** Track KPIs (ACoS, ROAS, conversion rates, keyword rankings), conduct competitive analysis, and use insights to refine strategies and maximize profitability.
- **Team Collaboration:** Work cross-functionally with creative, marketing, and supply chain teams to align product launches, content creation, and seasonal planning.
- **Compliance & Optimization:** Ensure all product listings meet Amazon's guidelines, including accurate categorization, high-quality images, and compelling, SEO-optimized descriptions.

**Qualifications**

- 5+ years of experience in Amazon operations or e-commerce management, with at least 2 years focused on apparel (specifically women's clothing).
- Proven success in scaling products to top 10,000 Amazon Best Seller Rank (BSR) in the Clothing category.
- Experience managing catalogs with fewer than 5,000 SKUs, with a focus on efficiency and growth.

- Advanced proficiency in Amazon Seller Central, along with tools like Helium 10, Jungle Scout, or SimilarWeb.
- Exceptional written and verbal English skills; ability to craft persuasive product listings and communicate effectively with stakeholders.
- Strong analytical mindset with expertise in interpreting sales data, market trends, and customer feedback.
- Highly organized, proactive, and capable of managing multiple priorities in a fast-paced environment.

**Preferred Qualifications**

- Experience with Amazon Ads campaign management (Sponsored Brands Video, Stores, etc.).
- Knowledge of U.S. fashion trends, sizing standards, and customer preferences.
- Background in logistics or supply chain management for apparel.
- Bachelor's degree in Business, Marketing, or a related field.

**What We Offer**

- Competitive salary and performance-based bonuses.
- Fully remote work flexibility with a collaborative, growth-oriented team.
- Professional development opportunities and access to cutting-edge e-commerce tools.
- A dynamic environment where your expertise will directly impact the company's growth and success.

**Application Process**
Please submit your resume and a cover letter detailing your relevant experience to [email address] with the subject line: *Senior Amazon Operations Manager Application – [Your Name]*. Applications will be reviewed on a rolling basis until the position is filled.

Join us in shaping the future of our brand and driving the next phase of growth on Amazon's most competitive shelves!

---

**Query**

如果人类社会中的人全部被猫或猫的拟人化形态所替代，请用500字来简单描绘一下它们的生活。

---

**LongWriter-Zero Think**

用户让我写一篇500字的短文，描述如果人类社会被猫或猫的拟人化形态替代后的世界。首先，我需要明确这个设定的核心：猫族取代人类，这意味着要将猫的习性、行为和生态特征融入人类社会结构中。用户没有具体说明是所有猫还是某类猫，所以可能需要泛指"猫族"，但要注意区分不同品种的猫在拟人化后的差异，比如虎斑猫、暹罗猫、布偶猫等，可能有不同的社会分工。

接下来要考虑受众和目的。用户可能对奇幻或科幻题材感兴趣，所以风格要轻松幽默，带点荒诞感，同时保持细节丰富。受众可能是喜欢幻想文学的读者，或者对动物拟人化有好奇心的人。目的可能是展示一个反乌托邦或乌托邦式的未来，突出猫族与人类社会的不同之处。

然后是结构设计。500字不长，需要紧凑的段落。开头可以描绘一个典型场景，比如上班时间，展现猫族的日常活动。中间分几个部分：工作、娱乐、家庭生活、科技与社会制度，每个部分用猫的特征来对应人类社会的元素。结尾可以展望未来，或者留下一个悬念，让读者想象更多细节。

风格方面，要保持活泼，多用比喻和拟人化的动作描写，比如"慵懒地伸展爪子"、"尾巴尖划过键盘"等，增强画面感。语气上可以带点揶揄，比如讽刺人类社会的某些现象，通过猫的视角来反观，增加趣味性。

受众适配方面，需要确保即使没有养猫经验的读者也能理解。可能需要解释一些猫的行为，比如蹭腿、打呼噜等，但不要过于直白，保持微妙的幽默感。同时，要注意避免种族歧视或刻板印象，确保所有猫族都具有多样性，比如有的勤奋，有的懒散，有的擅长科技，有的热爱艺术。

可能还需要反思用户是否希望突出某些主题，比如科技与自然的平衡、社会结构的转变、人类价值观的反思等。如果用户没有明确要求，可以适度融入这些元素，但不作为主线，保持故事的流畅性。

质量检查方面，要确保每个段落之间有自然的过渡，逻辑连贯。检查是否有重复的描述，比如多次提到"懒散"或"慵懒"，需要替换词汇。同时，注意字数控制，500字左右，可能需要删减次要细节，保留核心亮点。

最后，可能需要考虑是否加入一些冲突元素，比如不同品种猫之间的矛盾，或者与人类遗留问题的互动，但用户要求简单描述，所以可能不需要，保持整体轻松的基调即可。总之，要将猫的可爱与人类社会的结构结合，创造一个既有趣又独特的幻想世界。

## LongWriter-Zero Response

晨曦破开第一道光，整个城市才真正苏醒。街道两旁的路灯都换成猫薄荷形状，散发着诱人的清香。每个人（或者说每只猫）清晨的仪式是顺着墙根仔细嗅闻，然后在某处顺滑的墙面留下油亮的爪印——这是它们互通日常的重要方式，谁家的猫薄荷快吃完了，谁要举办抓毛线球派对，全在这些精心踩出的符号里。

通勤工具是时速五公里的猫车，喵妹们戴着猫铃坐好后，老司机（或说猫公）熟练地将车开到高处的停车场。办公场所是层层叠叠的阁楼，最顶层的会议桌是特制的猫爬架，当重要议题商讨到一半时，常有与会者突然一跃而上，全然忘记自己正穿着定制的高领毛衣。会议记录员负责把这种跃动纳入文件，毕竟真正有效的决策总是在攀爬间达成。

家庭生活更符合毛茸茸的本性。猫咪们晚上八点准时围坐在毛毯围成的"餐桌"旁，吃着猫罐头和鱼干拌饭，交流一天的见闻——用的是人类听不懂的"喵-喵-汪"混合语，只有猫耳一族的祖传翻译能破译。饭后最温馨的时刻是选几根毛线球，举着荧光棒在客厅里追逐光点，等到灯微微倾斜，全家人会一起半躺在沙发上，用头顶的肉垫互相按摩，直到管家喵叫着催促："去洗澡啦，地板可别留毛球！"

政府广场上，猫头鹰议员正在宣读法案："为维护城市美观，禁止在图书馆二楼以上抓挠墙壁。"但这些话在午后的阳光下显得有些无力，大家更关心的是新任猫管局长承诺的免费猫薄荷服务何时开始。学校里，老师教育小猫们的重点是"探索欲"——当历史课讲到古埃及猫王，老师会带它们去解剖猫乃伊的骨骼；数学课则教授如何用爪子计算猫粮的剩余量。

最特别的是社交货币——猫砂。有钱人家会在客厅铺上进口水晶猫砂，吸引全城猫咪来蹭蹭，而普通家庭则会收集干净的猫砂团，用作交换服务的筹码。垃圾回收日，铲屎官们需要在社区中心用狗饼干换取猫砂，这导致某些街区形成了"猫砂囤积户"，他们会用墙角的头像挂件警示邻居："请勿踏入，此处为私人猫砂仓库。"

夜幕降临时，城市进入睡眠模式，但每栋楼的天台都亮起激光灯，高处的猫咪们顺着光柱追逐到深夜，喵叫此起彼伏。这大概就是毛茸茸城市的秘密：它们用柔软的爪子重塑世界，让每一分存在都充满尊严，每一种生活都裹着温暖的绒毛。

## Deepseek-R1 Response

当第一缕晨光掠过尖顶钟楼时，整座城市开始舒展柔软的曲线。高耸的猫爬架建筑群表面覆盖着磨砂水晶板，折射出粼粼金斑，街道上浮动着鱼干与木天蓼混合的香气。裹着丝绸长袍的猫居民们竖起尾巴，瞳孔缩成细线，踩着肉垫在错落有致的空中回廊间跳跃。他们的足音被吸音材料吞噬，整座城市如同播放着默片的毛线球剧场。

市政厅穹顶悬挂着巨型激光逗猫棒，全息投影的红色光点精准落在每个工作日的九点整。猫公务员们用带有倒刺的舌头整理好文件，将带金枪鱼印章的文书塞进树洞状传输管道。交易所里的期货交易员盯着不断跳动的鲜鱼指数，爪子在触摸屏上划出残影，尾巴却诚实地拍打着真皮座椅——这个月鲭鱼期货的波动显然让某些人按捺不住本能。

商业区的橱窗陈列着镶钻项圈与鸵鸟毛玩具，戴着单片眼镜的猫掌柜趴在流线型柜台后假寐，尾巴尖却有节奏地敲击着隐藏的报警按钮。转角处的露天茶馆里，

穿着和服的老猫正用爪钩轻拨三味线，学徒端着抹茶羊羹穿梭时，总会被突然弹起的尾巴绊个趔趄。

黄昏时分，霓虹灯管缠绕的广告塔亮起「猫薄荷特供」的荧光字，穿铆钉皮衣的年轻猫们聚集在地下俱乐部，用改装过的电动梳毛器制造出震耳欲聋的嗡鸣。戴着夜视仪的巡逻队蹲踞在屋顶，突然集体炸毛——某栋公寓32层阳台上，哲学家猫又打翻了银河系投影仪，整个街区的光影瞬间扭曲成万花筒。

当月光为城市镀上银膜，所有建筑外立面的抓挠板自动弹出，亿万道猫爪划痕在夜色里泛起幽蓝磷光。这个瞬间，整座城市变成巨大的信息素留声机，记录着猫文明特有的、既慵懒又精确的生命韵律。

