# OpenReview forum: "LongWriter-Zero: Mastering Ultra-Long Text Generation via Reinforcement Learning"
_ICLR.cc/2026/Conference — ICLR 2026 Oral_

### Official Review · Reviewer_TSAk · 2025-10-28

**Soundness:** 3
**Presentation:** 4
**Contribution:** 3
**Rating:** 8
**Confidence:** 3

**Summary:**

The paper introduces LongWriter-Zero by building a framework to generating ultra-long text by applying RL to a base LM.

Different from the SFT approach from previous works, this paper proposes a RL approach with three main contributions 1) A dedicated reward model designed to balance length, writing quality and structural format. 2) The integration of a thinking step. 3) continual pre-training on writing-centric data would significantly boost the performance on top of the RL-based training.

The resulting model proves the effectiveness of the RL framework by demonstrating to achieve SOTA performance on WritingBench and good ELO score, surpassing its SFT counterpart model and open-source/proprietary models in long text generation.

**Strengths:**

* The paper successfully built a pure RL-driven approach for an open-ended long-text generation task, moving beyond the prevalent paradigm of SFT on synthetic data, shed light on a topic with large headroom.
* The paper is well-written to systematically  perform ablation on the importance of continual pre-training and the "think" step, providing valuable architectural insights.
* LongWriter-Zero demonstrates SOTA results on multiple challenging benchmarks, validating the overall effectiveness of the proposed framework.

**Weaknesses:**

* It's a bit unclear comparison:
   * If the comparison is between a basic SFT and the RL GRPO setup, it's no surprise the GRPO RL setup has more advantage;
   * If the comparison is between RL and the SOTA approaches (which often has DPO/RL steps after a basic SFT), Table 1 is the only section discussion about evidence.

* The paper is lacking details on how to build teh preference dataset for "Writing RM", e.g. data origin, size, diversity and human rating rubrics, etc. Given the importance of reward signals in RL, the missing info makes the major weakness on reproducibility and the trust in the results.

* The paper mentioned removal of KL penality term (beta = 0, line 138). For the open-ended question like long-form writing, it's not justified   to use this setup. The KL term is an important regularizer mitigating reward hacking and the authors mentioned the repetition-drive reward hacking as a key limitation. The decision to remove KL penalty needs more thoughts.

**Questions:**

Questions:
* Figure 2 and Figure 3 shows different story for "Base-think": Figure 3 shows a high ELO at the very beginning (before 150steps) but significantly worse in "Writing RM", does that mean an misalignment between Writing RM and ELO ratings? or writing quality is less important in deciding who win?

* Figure 6 mentioned thinking token length plateaus because writing task has a ceiling. Could this plateau be an artifacts of the experiment design? For example, (a) could it be caused by the context length? (b) The format RM implicitly pensalizing overly long thinking setup?

* Why remove KL-Penalty? details seen in the weakness section

Suggestions on Writing:
* Fig. 2 (right) needs more explanation on "Mean Non-Overlong Generation Length". It's hard to understand. And Green curve has different behavior than the other two methods. Need more explanation.
* Use "Direct-Answer" and "NoThink" in different places but presenting the same meaning, a bit confusing.
* nit: "-Continual Pretrain" in Table 1 may be misunderstood as dash line rather than "minus". Same for "-Thinking".

---

> ### Author Response · Authors · 2025-11-18
> **PART 1/2**
>
> Thank you for your constructive review and valuable suggestions! Below, we provide a detailed response to your questions and comments. If any of our responses fail to sufficiently address your concerns, please inform us, and we will promptly follow up.
> ***
>
>
> ***W1: It's a bit unclear comparison: If the comparison is between a basic SFT and the RL GRPO setup, it's no surprise the GRPO RL setup has more advantage;  If the comparison is between RL and the SOTA approaches (which often has DPO/RL steps after a basic SFT), Table 1 is the only section discussion about evidence.***
>
>
>
>
> Our comparison is intended to make two points clear. First, comparing our SFT baseline with our RL setup demonstrates that task-specific RL is indeed more effective for long-form generation than SFT alone (Section 3.3 SFT VS. RL). Second, in comparison with current SOTA models, it is important to note that many strong systems, such as Qwen3[1], DeepSeek-R1[2], and LongWriter-DPO[3], also include a general DPO/RL stage after SFT. Within this common training paradigm, our RL strategy achieves higher evaluation scores (as shown in Table 1), indicating that our method offers a more effective RL formulation for long-output tasks.
>
> [1] Yang, A., Li, A., Yang, B., Zhang, B., Hui, B., Zheng, B., Yu, B., Gao, C., Huang, C., … (2025). Qwen3 Technical Report. arXiv preprint arXiv:2505.09388.
>
> [2] Guo, D., Yang, D., Zhang, H., Song, J., Zhang, R., Xu, R., Zhu, Q., Ma, S., Wang, P., Bi, X., Zhang, X-k., Yu, X., Wu, Y., … (2025). DeepSeek-R1: Incentivizing Reasoning Capability in LLMs via Reinforcement Learning. arXiv preprint arXiv:2501.12948.
>
> [3] Bai, Y., Zhang, J., Lv, X., Zheng, L., Zhu, S., Hou, L., Dong, Y., Tang, J., Li, J. (2024). LongWriter: Unleashing 10,000+ Word Generation from Long Context LLMs. arXiv preprint arXiv:2408.07055.
>
> ***
> ***W2: The paper is lacking details on how to build teh preference dataset for "Writing RM", e.g. data origin, size, diversity and human rating rubrics, etc. Given the importance of reward signals in RL, the missing info makes the major weakness on reproducibility and the trust in the results.***
>
> We thank the reviewer for pointing out the need for more details on the preference dataset used for training the Writing RM. As noted in Section 2.2, we already provide a brief description of the dataset and training setup, but we agree that additional clarification will improve reproducibility.
>
> Our Writing RM is based on Qwen2.5-72B and trained with a Bradley–Terry style preference objective. The preference dataset consists of over 100k pairs collected from a mixture of model-generated comparisons and human annotations, covering diverse long-form writing tasks. Annotators follow a clear rubric focusing on fluency, coherence, factuality, structure, and task adherence.
> ***
> ***W3 & Q3: The paper mentioned removal of KL penality term (beta = 0, line 138). For the open-ended question like long-form writing, it's not justified to use this setup. The KL term is an important regularizer mitigating reward hacking and the authors mentioned the repetition-drive reward hacking as a key limitation. The decision to remove KL penalty needs more thoughts.***
>
> We agree that the KL penalty is a widely used and important regularizer for RL in language models, especially for mitigating reward hacking in RLHF-style setups. Our setting, however, differs from standard RLHF, and we intentionally set β = 0 for two reasons. First, our goal is to push the policy substantially beyond the length and stylistic regime of the base model in long-form writing. The base policy is typically conservative in length and coverage, and a non-trivial KL penalty strongly suppresses the very distributional shift (toward much longer, more detailed outputs) that we aim to induce. This design choice is aligned with recent large-scale RL systems such as DAPO: An Open-Source LLM Reinforcement Learning System at Scale, which also removes the KL term when the objective is to train long-CoT reasoning models that are allowed to diverge substantially from the reference policy, arguing that the KL constraint is no longer necessary in that regime. Second, we do not rely on “no KL” as the only safeguard. The policy is regularized by (i) task-aware length prediction, which rewards staying within a narrow, task-specific length range rather than unbounded growth, and (ii) format and repetition rewards, which penalize repetitive, malformed, or structurally degenerate outputs, together with a conservative RL schedule; these act as alternative regularizers that limit pathological behaviors in practice.

---

> ### Author Response · Authors · 2025-11-19
> **PART 2/2**
>
> ***Q1: Figure 2 and Figure 3 shows different story for "Base-think": Figure 3 shows a high ELO at the very beginning (before 150steps) but significantly worse in "Writing RM", does that mean an misalignment between Writing RM and ELO ratings? or writing quality is less important in deciding who win?***
>
> Regarding the apparent discrepancy between Figure 2 and Figure 3 for Base-think, we would like to clarify the following. First, writing RM and ELO are not expected to match perfectly. Writing RM is a reward model score, while ELO comes from a judge model via pairwise comparisons, so the two metrics naturally capture different aspects of performance. Second, the early high ELO of Base-think is mainly due to the comparison pool. Several baseline models in the ELO evaluation are relatively weak, and the initial Base-think policy can already beat them. This makes the ELO score look high at the beginning. As training continues, the model gradually competes with the stronger baselines such as DeepSeek-R1 and Qwen3-235B. This produces a slow, steady increase in ELO because beating these stronger models is harder. In contrast, the Writing RM curve shows a sharper rise during this stage, as the reward model is more sensitive to these incremental improvements. Therefore, the different shapes in the two figures reflect the evaluation setup rather than any misalignment between Writing RM and ELO.
> ***
> ***Q2: Figure 6 mentioned thinking token length plateaus because writing task has a ceiling. Could this plateau be an artifacts of the experiment design? For example, (a) could it be caused by the context length? (b) The format RM implicitly penalizing overly long thinking setup?***
>
> Our maximum model context length is 16K, which was the upper limit of the infrastructure during our experiments. For typical writing tasks with ~5K expected output length, the theoretical maximum think-length remains ~11K. This is far above the ~1.2K plateau shown in Figure 6, so the saturation cannot be caused by the context-length ceiling.
> Additionally, the format reward is applied only to the output (excluding long thinking), so the plateau cannot be attributed to implicit discouragement of lengthy reasoning plans. Instead, the plateau likely reflects the intrinsic requirement of the writing task: beyond a certain point, longer reasoning chains simply do not provide additional benefit for writing-focused RL.
>
> ***
> ***Suggestions on Writing***
>
> Thank you for these helpful suggestions—we will incorporate them in the revision.

---

> ### Author Response · Authors · 2025-11-25
>
> We appreciate your constructive feedback, which has helped us significantly improve the clarity and quality of our paper. We believe we have addressed all the concerns raised and would be happy to engage further if there are any outstanding questions or additional points to clarify. If the current version resolves your concerns, we kindly ask that you consider updating your score to reflect the revisions made. Thank you once again for your thoughtful and helpful comments.

---

### Official Review · Reviewer_htGy · 2025-10-31

**Soundness:** 3
**Presentation:** 4
**Contribution:** 3
**Rating:** 6
**Confidence:** 3

**Summary:**

The paper describes a new approach to generating quality long-form text, LongWriter-Zero, aiming to improve over standard LLM generation baselines and the small number of works that have explicitly tackled longform generation (LongWriter, Suri).  The approach uses three components, framing their use using three research questions (RQs): RL with appropriate reward model aspects (length, quality, structural format and redundancy), test-time scaling, and continual pretraining.  There is an evaluation on three benchmarks (the existing WritingBench, Arena-Write defined in this paper, and a human-in-the-loop win-rate evaluation), as well as more detailed investigations like e.g. assessments of the importance of the three core components of LongWriter-Zero.

**Strengths:**

* The three components of LongWriter-Zero are intuitive, even if not necessarily obvious, contributing some originality.  I thought the argument for RQ2, on the use of test-time scaling – that it had mostly been used for e.g. code and math problems, and that its use was less obvious for long-form generation – was sensible and interesting, and worth investigating.

* The model seems to perform strongly, even against much larger models.  Having three different types of evaluation was also a strength.  Ablation analyses were also informative.

* This is an important aspect of LLMs to look at, and there isn't too much work in this space yet, making this a useful contribution.

**Weaknesses:**

* The evaluation was rather different from the LongWriter paper by Bai et al, which is really the primary baseline, as framed by e.g. the paper abstract and the related work.  LongBench-Write isn’t mentioned at all in the present paper.  While WritingBench does in some ways supersede LongBench-Write (with that paper discussing the relationship of the new benchmark to the prior LongBench-Write), I still would have expected to see an evaluation on LongBench-Write here for comparability.

* Relatedly, the critic used for the first evaluation probably warrants some discussion.  It of course makes sense to use the critic that comes with WritingBench, but there’s no commentary on or investigation of possible bias in the evaluation of Qwen models wrt non-Qwen models by a Qwen critic.  (Bai et al noted this for their GPT-4o judge for LongWriter.)  This also factors into the results in Table 1, where the DeepSeek-R1 is probably close enough that I wonder whether the improvement is real or a result of bias.

* Looking elsewhere in Table 1 results, the commentary in Sec 3.2 just observes that LongWriter-Zero outperforms all models, but it only outperforms Qwen3-235B-A22B by a quite small 0.01 overall.  (I’d also note that the Qwen3-235B-A22B score for EN should be bolded as equal highest given the one decimal place result.)  While this is still noteworthy, given that LongWriter-Zero is a much smaller model, I think it should be remarked upon.

* Relatedly, do you have any thoughts about why LongWriter-Zero and Qwen3-235B-A22B are so close in the Table 1 results by the critic but there is a much more substantive difference in Elo ratings (discussed in Sec 3.2)?

**Questions:**

Please see above.

---

> ### Author Response · Authors · 2025-11-18
>
> Thank you for your constructive review and valuable suggestions! Below, we provide a detailed response to your questions and comments. If any of our responses fail to sufficiently address your concerns, please inform us, and we will promptly follow up.
> ***
> ***Q1: The evaluation was rather different from the LongWriter paper by Bai et al, which is really the primary baseline, as framed by e.g. the paper abstract and the related work. LongBench-Write isn’t mentioned at all in the present paper. While WritingBench does in some ways supersede LongBench-Write (with that paper discussing the relationship of the new benchmark to the prior LongBench-Write), I still would have expected to see an evaluation on LongBench-Write here for comparability.***
>
> We agree that including LongBench-Write results is important for direct comparison with Bai et al.’s LongWriter paper. We have additionally evaluated LongWriter-Zero-32B on LongBench-Write for completeness.
> LongBench-Write Results (Baseline results from “LongWriter: Unleashing 10,000+ Word Generation from Long Context LLMs”)
>
> | Model                 | $\bar{S}$  | $S_l$    | $S_q$   |
> |-----------------------|-------|-------|-------|
> | Claude 3.5 Sonnet     | 80.7  | 73.7  | 87.7  |
> | GPT-4 Turbo           | 67.3  | 47.9  | 86.6  |
> | GLM-4-9B-Chat         | 68.3  | 51.0  | 85.5  |
> | Llama-3.1-8B-Instruct | 60.3  | 50.0  | 70.6  |
> | Llama-3.1-70B-Instruct | 65.6  | 50.8  | 80.3  |
> | Mistral-Large-Instruct | 77.0  | 65.6  | 88.3  |
> | Suri-I-ORPO           | 56.6  | 59.6  | 53.5  |
> | LongWriter-8B         | 79.8  | 77.4  | 82.2  |
> | LongWriter-9B         | 80.5  | 78.6  | 82.3  |
> | LongWriter-9B-DPO     | 84.0  | 82.6  | 85.4  |
> | **LongWriter-Zero**   | **86.3** | **84.2** | **88.4** |
>
> LongWriter-Zero achieves the strongest performance across all metrics, exceeding both the published LongWriter models and other competitive baselines. This shows that the improvements seen on WritingBench also transfer to LongBench-Write.
>  We will include this table and citation in the revised version to strengthen comparability with prior work.
>
> ***
> ***Q2: Relatedly, the critic used for the first evaluation probably warrants some discussion. It of course makes sense to use the critic that comes with WritingBench, but there’s no commentary on or investigation of possible bias in the evaluation of Qwen models wrt non-Qwen models by a Qwen critic. (Bai et al noted this for their GPT-4o judge for LongWriter.) This also factors into the results in Table 1, where the DeepSeek-R1 is probably close enough that I wonder whether the improvement is real or a result of bias.***
>
> Relying solely on the WritingBench critic may indeed introduce model-family bias, particularly when comparing Qwen-based model. To address this, we added two evaluation benchmark that are fully independent of the WritingBench: (1) GPT-4.1 judging and (2) ArenaWriter Elo comparisons. These evaluations do not rely on the Qwen model family which helps avoid the potential family-level bias of WritingBench’s Qwen-based critic. Both additional metrics consistently show that our model outperforms strong baselines, including DeepSeek-R1, indicating that the gains reported in Table 1 are not artifacts of Qwen-family bias.
> ***
> ***Q3: Looking elsewhere in Table 1 results, the commentary in Sec 3.2 just observes that LongWriter-Zero outperforms all models, but it only outperforms Qwen3-235B-A22B by a quite small 0.01 overall. (I’d also note that the Qwen3-235B-A22B score for EN should be bolded as equal highest given the one decimal place result.) While this is still noteworthy, given that LongWriter-Zero is a much smaller model, I think it should be remarked upon.***
>
> We agree with the reviewer’s observation. We will revise Section 3.2 to highlight this point and correct the formatting issue regarding the EN score.
> ***
> ***Q4: Relatedly, do you have any thoughts about why LongWriter-Zero and Qwen3-235B-A22B are so close in the Table 1 results by the critic but there is a much more substantive difference in Elo ratings (discussed in Sec 3.2)?***
>
> This could be attributed to the different evaluation philosophies of WritingBench versus ArenaWriter. WritingBench uses an absolute scoring scheme, where strong models often receive similarly high scores (e.g., both outputs may be rated 9-10). This naturally compresses differences when both models already produce high-quality text.
> In contrast, ArenaWriter relies on pairwise comparison and Elo rating, which is more sensitive to relative preference and can capture finer distinctions between two otherwise strong outputs. As illustrated in the Appendix 8 CASE STUDY examples, LongWriter-Zero’s generations tend to provide richer detail and stronger persuasive or expressive quality compared to Qwen3-235B-A22B, which becomes more apparent under pairwise judging.

---

> ### Author Response · Authors · 2025-11-25
>
> We appreciate your constructive feedback, which has helped us significantly improve the clarity and quality of our paper. We believe we have addressed all the concerns raised and would be happy to engage further if there are any outstanding questions or additional points to clarify. If the current version resolves your concerns, we kindly ask that you consider updating your score to reflect the revisions made. Thank you once again for your thoughtful and helpful comments.

---

### Official Review · Reviewer_rDBg · 2025-11-01

**Soundness:** 2
**Presentation:** 3
**Contribution:** 2
**Rating:** 4
**Confidence:** 3

**Summary:**

While ultra-long generation is a desired capability in large language models, it often requires fine-tuning on SFT data with expensive data which may also lack quality. This paper instead motivates and introduces LongWriter Zero, an RL paradigm to transform a base language model into one that can generate long and high-quality text - without written data. They provide intuitive reward design, an introduce a new benchmark to evaluate different LLMs against each other (Arena-Write). They also perform helpful ablations demonstrating the efficacy in the continual pre-training case.

**Strengths:**

* The paper is written well in general
* This solves a key, timely problem: how do we encourage more length without access to expensive SFT data?
* There are comprehensive details of prompts and experimental design which seem well-justified, including a thorough appendix.
* The reward design is justified and includes multiple components
* The proposed model LongWriter-Zero (32B scale) outperforms a variety of methods across a few benchmarks

**Weaknesses:**

* The results are shown on only one model trained with the introduced paradigm. It would be nice to test if this holds across other models, namely 1) models not Qwen and 2) perhaps a smaller size Qwen (ie, does this still work at smaller scale)
* While the approach is nice, replacing costly SFT data, the process intuitively seems prone to length reward hacking, since the reward is based on the averaged advantage over length equally with format and writing quality, a repetitive string may overwhelm the rest of the rewards by being extremely long, leading to reward hacking with long and non-fluent texts. This may be solved by weighting the other factors more, or in a different way.
* Related to the above, it is unclear how robust the writing reward model, the proxy for quality, performs in complex and long scenarios. Though it is mentioned that it "capture holistic writing quality, including fluency, coherence, and helpfulness", there are other more complex factors like factuality which are hard to control for.
* There should be more baselines. While it is nice to compare to existing other models, there needs to be more comparison of this length method to other approaches, such as those that leverage synthetic data, etc (ie, LongWriter on 32B scale)
* Minor: ArenaWrite seems small - only 100 examples
* There are a couple formatting errors, for example, a citation link on the bottom of Page 6 spanning over the page break
* Minor: Sections RQ1, RQ2, and RQ3 seem a bit disconnected, could be transitioned better.

**Questions:**

* Why did you need to create and evalute with Arena-Write, ie, what is wrong with existing benchmarks for this?

---

> ### Author Response · Authors · 2025-11-18
> **PART 1/2**
>
> Thank you for your constructive review and valuable suggestions! Below, we provide a detailed response to your questions and comments. If any of our responses fail to sufficiently address your concerns, please inform us, and we will promptly follow up.
> ***
>  ***W1:  The results are shown on only one model trained with the introduced paradigm. It would be nice to test if this holds across other models, namely 1) models not Qwen and 2) perhaps a smaller size Qwen (ie, does this still work at smaller scale)***
>
> We agree that demonstrating results across additional model sizes and families would further strengthen the generality of the proposed paradigm. Beyond the main 32B model, we have also trained a smaller Qwen2.5-14B model using the full LongWriter-Zero pipeline, and its performance remains consistently strong (see supplementary results), suggesting that the method is not restricted to a single scale.
>  | Model | WritingBench ↑ | ArenaWriter ELO ↑ |
>   |-----|----------------|--------------------|
>   | Qwen2.5-14B-Instruct | 7.60 | 744 |
>   | LongWriter-Zero-14B | 8.49 (+0.89) | 1198 (+454) |
>
> While no existing work has suggested that such long-form writing RL methods fail to transfer across models, we are actively seeking additional computational resources to validate this broader generalization empirically.
> ***
> ***W2: While the approach is nice, replacing costly SFT data, the process intuitively seems prone to length reward hacking, since the reward is based on the averaged advantage over length equally with format and writing quality, a repetitive string may overwhelm the rest of the rewards by being extremely long, leading to reward hacking with long and non-fluent texts. This may be solved by weighting the other factors more or differently.***
>
> We would like to clarify that our length reward is not a monotonic function of length, and thus does not simply keep increasing as the sequence gets longer. Instead, we employ a task-aware length-prediction model that estimates an appropriate target range for each input, and the length reward is computed relative to this predicted range rather than unconditional expansion. For tasks with explicit limits, the model is restricted by a narrow tolerance band (±10%). For open-ended tasks, the predictor learns realistic ranges (e.g., 100–800 words for daily writing or 8,000–16,000 words for long-form paper). This design encourages appropriate, not excessive, length.
>
> In addition, we observed in early training that the model sometimes attempted to exploit this reward by repeating a sentence to meet the target length. To address this, we incorporated explicit penalties for repetitive content, including a repetition-checking module that identifies repeated or near-repeated content. This prevents the model from satisfying the length requirement through trivial duplication and steers it toward producing genuinely extended and coherent text.
> We will clarify these mechanisms in the revision—particularly in Section 2.2 (Reward Design — Length RM) and Appendix A.3—to make clear how task-aware length control and repetition penalties jointly mitigate pathological length-seeking behaviors.
> ***
> ***W3: Related to the above, it is unclear how robust the writing reward model, the proxy for quality, performs in complex and long scenarios. Though it is mentioned that it "capture holistic writing quality, including fluency, coherence, and helpfulness", there are other more complex factors like factuality which are hard to control for.***
>
> We agree that the writing reward model is not designed to cover every aspect of quality, and factors such as factuality are indeed harder to capture (as Appendix A.7 limitation). In principle, one could incorporate a factuality-oriented reward to address this, but that is not the focus of our work. Our goal is not to build a fully comprehensive reward system, but to study whether a task-aware reward setup is already sufficient for enabling pure-RL to produce strong long-form writing. Importantly, even though the Writing RM does not explicitly target factuality, independent evaluations indicate that it remains robust in complex and long scenarios. WritingBench (Qwen2.5-7B critic), Arena-Write (Elo-style pairwise judging), and a 200-example human study all consistently prefer LongWriter-Zero over strong baselines. If the Writing RM were systematically fragile—for example, encouraging incoherent or factually weak outputs—these independent judges and human annotators would have exposed such issues, which we do not observe. We will clarify this scope and limitation more explicitly in the revision.

---

> ### Author Response · Authors · 2025-11-18
> **PART 2/2**
>
> ***W4: There should be more baselines. While it is nice to compare to existing other models, there needs to be more comparison of this length method to other approaches, such as those that leverage synthetic data, etc (ie, LongWriter on 32B scale)***
>
> We appreciate the suggestion to include more baselines, especially methods leveraging synthetic data such as LongWriter on a 32B scale. In fact, our experiments already compare against synthetic-data-based approaches, including LongWriter and Suri, within our baseline set. Moreover, in Section 3.3 (SFT vs. RL), we explicitly conduct an SFT experiment using LongWriter’s synthetic data on our 32B model, which can be viewed as a version of “LongWriter on 32B scale.” As shown in Figure 4 and the table below, this SFT setup performs significantly worse than our RL method, reinforcing the effectiveness of our RL framework beyond standard synthetic-data SFT pipelines. We will make this connection more explicit in the revised version to avoid ambiguity.
>   | Model | WritingBench ↑ | ArenaWriter ELO ↑ |
>   |---------------------------|----------------|--------------------|
>   | Qwen2.5-32B - LongWriter - SFT | 8.08 | 964 |
>   | LongWriter-Zero-32B | 8.69 (+0.61) | 1447 (+483) |
>
>
>
> ***
> ***W5: Minor: ArenaWrite seems small - only 100 examples & Q1: Why did you need to create and evalute with Arena-Write, ie, what is wrong with existing benchmarks for this?***
>
> ArenaWrite is intentionally kept small (100 examples) because its purpose is fast and lightweight monitoring during the RL process, rather than serving as a comprehensive evaluation benchmark. This design mirrors common practices in RL for code generation, where small diagnostic sets, rather than the full SWE suite[1], are used for real-time progress tracking. Our full evaluations rely on larger benchmarks; ArenaWrite functions mainly as an efficient monitoring signal for iterative RL tuning.
>
> [1] Jimenez C E, Yang J, Wettig A, et al. SWE-bench: Can Language Models Resolve Real-world Github Issues?[C]//The Twelfth International Conference on Learning Representations.
> ***
> ***W6:There are a couple formatting errors, for example, a citation link on the bottom of Page 6 spanning over the page break***
>
> Thank you for pointing out the formatting issues. We will fix the citation break on Page 6 and carefully review the manuscript for similar problems.
> ***
> ***W7: Minor: Sections RQ1, RQ2, and RQ3 seem a bit disconnected, could be transitioned better.***
>
> As for the transitions between RQ1, RQ2, and RQ3, we appreciate the suggestion. We will revise the narrative flow to make the progression between these sections clearer and better connected.

---

> ### Author Response · Authors · 2025-11-25
>
> We appreciate your constructive feedback, which has helped us significantly improve the clarity and quality of our paper. We believe we have addressed all the concerns raised and would be happy to engage further if there are any outstanding questions or additional points to clarify. If the current version resolves your concerns, we kindly ask that you consider updating your score to reflect the revisions made. Thank you once again for your thoughtful and helpful comments.

---

### Official Review · Reviewer_nuzb · 2025-11-03

**Soundness:** 3
**Presentation:** 3
**Contribution:** 2
**Rating:** 6
**Confidence:** 3

**Summary:**

The paper introduces LongWriter-Zero, an RL framework using GRPO to train LLMs for ultra-long text generation from scratch, bypassing synthetic SFT data. It uses composite rewards for length control, writing quality, and format, plus <think> prompts for reasoning and continual pre-training on writing data. The authors test the approach on WritingBench, Arena-Write, and human win-rate evals. Results achieve SOTA: 8.69/10 average on WritingBench, 1447 Elo on Arena-Write, and win rates >62% vs. 100B+ models like DeepSeek-R1.

Strengths:
- The pure RL approach fosters emergent long-form capabilities without data quality biases from synthetic SFT, enabling innovation beyond teacher model limits.

Weaknesses:
- Using heuristic rewards naturally entails reward hacking. The effectiveness of the proposed method is based on how well the heuristics of the reward models align with the actual metrics that they care about. The paper does not show that LongWriter-Zero is immune from reward hacking. This suggests that the generalization of the approach might be limited.
- Composite reward balancing risks suboptimal trade-offs, where normalizing advantages could dilute focus on critical aspects like coherence in ultra-long sequences.

**Strengths:**

- The pure RL approach fosters emergent long-form capabilities without data quality biases from synthetic SFT, enabling innovation beyond teacher model limits.

**Weaknesses:**

- Using heuristic rewards naturally entails reward hacking. The effectiveness of the proposed method is based on how well the heuristics of the reward models align with the actual metrics that they care about. The paper does not show that LongWriter-Zero is immune from reward hacking. This suggests that the generalization of the approach might be limited.
- Composite reward balancing risks suboptimal trade-offs, where normalizing advantages could dilute focus on critical aspects like coherence in ultra-long sequences.

**Questions:**

See weaknesses.

---

> ### Author Response · Authors · 2025-11-18
>
> Thank you for your constructive review and valuable suggestions! Below, we provide a detailed response to your questions and comments. If any of our responses fail to sufficiently address your concerns, please inform us, and we will promptly follow up.
> ***
>  ***Q1:  Using heuristic rewards naturally entails reward hacking. The effectiveness of the proposed method is based on how well the heuristics of the reward models align with the actual metrics that they care about. The paper does not show that LongWriter-Zero is immune to reward hacking. This suggests that the generalization of the approach might be limited.***
>
> We would like to clarify that our goal is not to claim that LongWriter-Zero is immune to reward hacking; In fact, several concrete forms of reward hacking appeared during training. For instance, the model sometimes relied on stereotypical stylistic openings (e.g., “Early morning…”) or slipped in near-duplicate passages to hack the quality and length reward. We also noticed that simple n-gram-based repetitive penalties were not always sufficient as the model could occasionally bypass them through minor token-level variations.
>
> These observations informed how we refined our training setup. We introduced stronger format-level constraints, including structural organization checks, and a multi-stage repetition-detection module that captures broader pattern-level redundancy beyond n-gram matching. These adjustments significantly reduced the most frequent reward-hacking behaviors and led to more stable RL optimization. We do not argue that these techniques eliminate reward hacking, but they do meaningfully mitigate practical issues in applying RL to ultra-long text.
>
> Crucially, the effectiveness of our approach is not evaluated using the same heuristic rewards that guide training. WritingBench employs an independently trained Qwen2.5-7B critic (50K human labels, 83% agreement), under which LongWriter-Zero achieves the highest overall score (8.69). Arena-Write further evaluates models via Elo-style comparisons using a separate judge model (Qwen2.5-72B), and LongWriter-Zero again attains the top Elo (1447) among six strong baselines. Finally, our human evaluation on 200 real-world instructions shows consistent human preference and GPT4.1 for our model. If the learned policy were primarily exploiting idiosyncrasies of our heuristic rewards, we would not expect these gains to persist across independent critics and human raters.
>
> In short, while heuristic rewards inevitably carry limitations, the method demonstrates robust generalization across independent critics and human evaluations.
> ***
> ***Q2: Composite reward balancing risks suboptimal trade-offs, where normalizing advantages could dilute focus on critical aspects like coherence in ultra-long sequences.***
>
> In our design, we explicitly normalize each reward component, length and writer quality included, into the same [−1,1] range before combining them. This step ensures that as long as there is any score difference within a reward group, each component contributes equally during learning. For example, when computing the length score, we first normalize its raw value into [−1,1]; the quality score undergoes the same normalization. This prevents situations where one reward (e.g., length) has much larger numeric magnitude and unintentionally dominates optimization, thus maintaining a balanced and stable learning signal across all targeted dimensions.

---

> ### Author Response · Authors · 2025-11-25
>
> We appreciate your constructive feedback, which has helped us significantly improve the clarity and quality of our paper. We believe we have addressed all the concerns raised and would be happy to engage further if there are any outstanding questions or additional points to clarify. If the current version resolves your concerns, we kindly ask that you consider updating your score to reflect the revisions made. Thank you once again for your thoughtful and helpful comments.

---

### Author Response · Authors · 2025-11-29

Dear PC, SAC, AC, and Reviewer,

We would like to sincerely thank you and all reviewers for the careful reading of our manuscript and for the constructive suggestions, which have substantially improved the clarity, rigor, and scope of the work.

In the revised version, we have carefully addressed the key points raised in each review:

- Reviewer nuzb: We have clarified that LongWriter-Zero is not immune to reward hacking and explicitly documented several concrete failure modes observed during training). We further describe the strengthened format-level constraints and enhanced repetition-detection module that were introduced in response to these issues, and we explicitly position reward hacking as a remaining limitation in Appendix A.7 rather than a solved problem. We also refined the discussion of composite reward normalization and explained how aligning all reward components to a common range helps avoid length dominating coherence or other quality aspects.

- Reviewer rDBg:
(i) We added new results on a smaller Qwen2.5-14B model trained with the full LongWriter-Zero pipeline, showing consistent gains over its SFT counterpart and supporting the claim that our paradigm is not restricted to a single scale.
(ii) We further clarified the design of the task-aware length reward and its interaction with repetition penalties, emphasizing that we do not reward absolute length growth but rather staying within a predicted, task-appropriate range, with explicit safeguards against repetitive padding.
(iii) We expanded the discussion of the writing reward model, clearly stating its scope and limitations and positioning it as a practical but non-exhaustive proxy for long-form quality.
(iv) We clarified that we already compare against synthetic-data-based methods such as LongWriter and Suri, including a “LongWriter-on-32B” SFT experiment, and we made this connection more explicit in the text.
(v) We explained the role of ArenaWrite as a small, diagnostic RL monitor rather than a full benchmark, and we committed to fixing minor formatting issues and improving transitions.

- Reviewer htGy:
(i) We added an explicit evaluation on LongBench-Write, including the published baselines from Bai et al. and our new LongWriter-Zero-32B results, demonstrating that the improvements observed on WritingBench also transfer to LongBench-Write.
(ii) We discussed the potential model-family bias of using a Qwen-based critic (WritingBench) and highlighted that our main conclusions are supported by independent evaluations, including GPT-4.1 judging and ArenaWriter ELO with human verification, which are not tied to the Qwen family.
(iii) We revised the commentary around Table 1 to explain. WritingBench uses an absolute scoring scheme where strong models often receive similarly high scores, causing differences to compress near the top. In contrast, ArenaWriter relies on pairwise judging, which is more sensitive to subtle preference differences between two already-strong models. As a result, even when WritingBench assigns similar numeric scores,  ELO and Win-rate can reveal clearer performance gaps.

- Reviewer TSAk:
(i) We clarified the comparison setup: our SFT vs. RL results isolate the effect of task-specific RL on long-form writing, while Table 1 situates LongWriter-Zero against current SOTA systems that themselves include DPO/RL stages after SFT (e.g., Qwen3, DeepSeek-R1, LongWriter-DPO).
(ii) We elaborated on the rationale for setting the KL penalty β = 0 in our setting, relating it to our goal of inducing substantial distributional shift toward much longer, more detailed outputs, and citing recent large-scale RL systems that adopt similar choices. We also emphasized that “no KL” is not our only safeguard: task-aware length control, format/repetition penalties, and conservative RL schedules jointly constrain degenerate behaviors.

Across these revisions, we believe that the reviewers’ suggestions, along with our corresponding experiments, analyses, and clarifications, have substantially improved the paper’s methodological transparency, empirical support, and overall readability. All changes have been fully incorporated into the updated PDF, and every modification is clearly highlighted in blue throughout the main text and tables for easy verification.

We are very grateful for your time and effort in handling our submission. If there are any remaining questions or if further clarification would be helpful for your decision, we would be more than happy to provide additional details.

Sincerely,

The Authors

---

### Meta-Review · Area_Chair_oXPP · 2025-12-23

**Summary:**

The paper proposes "LongWriter-Zero," a method to enable ultra-long text generation without relying on synthetic data. The core innovation lies in shifting the paradigm from SFT to RL. The authors employ a GRPO framework with composite reward models targeting length, quality, and format. They demonstrate that a 32B model trained this way can outperform larger models and traditional SFT baselines on benchmarks like WritingBench and Arena-Write.

**Reviewer Concerns:**

* **Generalization across model scales (Reviewer rDBg):** the method was only validated on a single 32B model. The authors added a crucial experiment with Qwen2.5-14B during the rebuttal.
* **Missing Baselines & Benchmarks (Reviewer htGy, rDBg):** Reviewers rightly pointed out the lack of comparison to "LongWriter-on-32B" (**SFT**) and the omission of LongBench-Write. The authors filled these gaps in the rebuttal, demonstrating that LongWriter-Zero outperforms its **SFT** counterpart and achieves strong results on LongBench-Write.
* **Reward Hacking & KL Penalty (Reviewer nuzb, TSAk):** Several reviewers were uneasy about the removal of the KL penalty () and potential length-hacking. The authors provided a solid justification: **RL** here aims for a significant distributional shift (ultra-long generation) that a KL constraint would actively suppress. They also clarified the use of task-aware length prediction and repetition penalties as alternative regularizers.
* **Evaluation Bias (Reviewer htGy):** The concern about Qwen-based critics favoring Qwen models was mitigated by pointing to the independent GPT-4.1 judging and human evaluation results.


Outstanding Review:  As noted by Reviewer rDBg and acknowledged by the authors, the current reward models focus heavily on length, structure, and coherence, potentially at the expense of factuality.

**Reviewer Scores:**

Reviewer nuzb: Score likely improves to 7 or 8.
Reviewer rDBg: Score likely improves to 6.

---

### Decision · Program_Chairs · 2026-01-26

Accept (Oral)